# Robustly federated learning model for identifying high-risk patients with post-operative gastric cancer recurrence

Bao Feng[1,2,7], Jiangfeng Shi[2,3,7], Liebin Huang[1,7], Zhiqi Yang[4], Shi-Ting Feng[5], Jianpeng Li[6], Qinxian Chen[1], Huimin Xue[1], Xiangguang Chen[4], Cuixia Wan[4], Qinghui Hu[2], Enming Cui [1], Yehang Chen [2] ✉ & Wansheng Long [1] ✉

The prediction of patient disease risk via computed tomography (CT) images and artificial intelligence techniques shows great potential. However, training a robust artificial intelligence model typically requires large-scale data support. In practice, the collection of medical data faces obstacles related to privacy protection. Therefore, the present study aims to establish a robust federated learning model to overcome the data island problem and identify high-risk patients with postoperative gastric cancer recurrence in a multi-centre, cross-institution setting, thereby enabling robust treatment with significant value. In the present study, we collect data from four independent medical institutions for experimentation. The robust federated learning model algorithm yields area under the receiver operating characteristic curve (AUC) values of 0.710, 0.798, 0.809, and 0.869 across four data centres. Additionally, the effectiveness of the algorithm is evaluated, and both adaptive and common features are identified through analysis.

Gastric cancer is one of the most prevalent malignancies[1]. Most patients with gastric cancer are diagnosed at an advanced stage[2]. Although surgical resection is regarded as the primary curative treatment for advanced gastric cancer (AGC), survival is unsatisfactory owing to the high incidence of recurrence after surgery[3,4]. Previous studies have shown that the Tumour Node Metastasis (TNM) staging system is the primary foundation for treatment planning and prognosis evaluation in AGC patients[5]. Unfortunately, even among patients with the same TNM stage, the clinical prognosis often varies. The TNM staging system lacks information on various tumour-related factors and tumour margin features, which are crucial for predicting postoperative recurrence in AGC patients[3,6,7]. Therefore, there is an urgent need to develop a method that can be used to identify high-risk patients prone to recurrence after curative gastrectomy, thereby

enabling early intervention (such as adjuvant chemotherapy and close follow-up) and improving patient outcomes.

In recent years, artificial intelligence (AI) technology has received widespread attention in the medical field and has shown exciting results[8-10]. However, a stable and effective AI-assisted diagnostic model relies not only on appropriate algorithms but also on large training datasets[11,12]. This large training dataset requires patient data to be shared across medical centres. At this time, medical organisations relinquish control of their own data. The security and privacy of patient data is difficult to protect, especially between countries, even creating a data monopoly situation. Therefore, multicentre data sharing is difficult to achieve.

Several researchers have proposed federated learning (FL), which trains a single shared model on a centre by aggregating local models

[1]Department of Radiology, Jiangmen Central Hospital, Jiangmen, China. [2]Laboratory of Intelligent Detection and Information Processing, Guilin University of Aerospace Technology, Guilin, China. [3]School of Electronic Engineering and Automation, Guilin University of Electronic Technology, Guilin, China. [4]Department of Radiology, Meizhou People's Hospital, Meizhou, China. [5]Department of Radiology, The First Affiliated Hospital of Sun Yat-sen University, Guangzhou, China. [6]Department of Radiology, Dongguan People's Hospital, Dongguan, China. [7]These authors contributed equally: Bao Feng, Jiangfeng Shi, Liebin Huang. ✉e-mail: cyh93yl@163.com; jmlws2@163.com

that are trained using only its own data from each medical centre. FL ensures data security while encouraging multicentre collaboration, which may lead to the development of more accurate and general AI-assisted diagnostic systems[13]. However, due to the differences in medical image collection equipment and regions at various medical centres, multicentre data may have problems with nonindependent and identically distributed data (such as different ratios of positive and negative data samples, as well as different distributions of image data)[14–16]. Thus, the shared model in federated learning is unable to meet the needs of all centres; for example, the performance of the shared model centre A is better, and the performance of the shared model in centre B is worse.

Driven by these real-world issues, we developed a robust federated learning model (RFLM) in the present study to accurately predict the risk of postoperative recurrence in patients with AGC. By effectively combining raw computed tomography (CT) image data from multiple centres while ensuring the privacy of individual patients, the present model significantly enhances predictive performance and generalisability without relying on centralised control over the final model. Moreover, we verified the effectiveness of the RFLM by evaluating the features extracted by the RFLM.

## Results

### The RFLM predicts postoperative recurrence in gastric cancer patients

The present study demonstrated that the proposed RFLM exhibited greater diagnostic efficiency when applied to data from the four centres. The area under the curve (AUC) values obtained from the test sets of the centres were 0.7101, 0.7981, 0.8091, and 0.8691. Comparison of the performance of the RFLM to that of the clinical model demonstrated an overall improvement in accuracy of 32.36% on the test dataset from all four centres. Furthermore, the RFLM successfully reduced the misdiagnosis rate of local recurrence of gastric cancer by 42.23% (Table S1).

The superior performance of the RFLM compared to that of the clinical model was likely attributed to its capacity to extract highly relevant common features that were closely linked to the specific task, which enabled the RFLM to effectively differentiate between images of local recurrence and nonrecurrence in gastric cancer patients. The integrated discrimination improvement (IDI) and net reclassification index (NRI) further confirmed that the RFLM significantly outperformed the clinical model on the data from each centre (Table S2).

### The RFLM outperforms other algorithms

To further assess the performance of the RFLM, a comparison with four traditional federated learning algorithms, namely, FedAvg[15], FedProx[16], Moon[17], and HarmoFL[18], as well as two robust federated learning algorithms, namely, pFedMe[14] and pFedFSL[19], was conducted.

Figure S1 displays the receiver operating characteristic (ROC) curves, while Fig. 1 illustrates the decision curve analysis (DCA) curves for all seven models using the data from the four centres. The RFLM consistently achieves the highest AUC results across all four centres (Table 1), with detailed diagnostic performance parameters presented in Table S3.

### The RFLM exhibits strong robustness

In deep learning, the performance of a model can be influenced by the dataset distribution. To evaluate the robustness of the proposed algorithm, we conducted five random permutations of the training dataset distribution across multiple centres, which demonstrated that the permutation of datasets had a minimal impact on the performance of the multicentre model. The AUC values of the four centres for the five experiments were $0.704 \pm 0.008$, $0.777 \pm 0.020$, $0.780 \pm 0.019$, and $0.836 \pm 0.009$ (Fig. S2).

To further verify the robustness of the model, the results of multicentre RFLM were verified by threefold cross-validation. The average AUC values for cross-validation across the four data centres were 0.722, 0.774 0.755, and 0.813 (Fig. 2).

To determine that the performance of the RFLM algorithm was independent of patient characteristics, such as sex and age, a stratified analysis was conducted to evaluate whether the performance of the algorithm significantly varied based on these factors. The performance of the RFLM algorithm was not affected by patient sex or age. These findings were confirmed by the DeLong test, which indicated that all $p$ values were greater than 0.05 (Fig. S3).

### The RFLM identifies both common and adaptive features across different data centres

To analyse the federated features of the RFLM, the common and adaptive features were examined. Common features refer to the category features shared by patients with and without gastric cancer recurrence, whereas adaptive features refer to the distinctive features specific to different data centres. To facilitate the demonstration of these joint features of common and adaptive features, 5 common features and 5 adaptive features were selected from 200 radiomic features at each data centre to generate joint feature heatmaps, providing illustrative representations of common and adaptive features across data centres.

Figure 3 shows the heatmaps of common features. In Centre A, features 1, 2, and 3 exhibited stronger correlations with features in the other three centres. Similarly, in Centre B, features 2, 3, and 5 exhibited stronger correlations with features in the other three centres. Centre C demonstrated stronger correlations between features 3, 4, and 5 and features in the other three centres. In Centre D, features 2, 3, and 4 displayed stronger correlations with features in the other three centres. In addition, examination of adaptive features indicated a strong correlation among the five features within each of the four data centres. However, the correlation with features from the other groups was relatively weaker. The federated feature heatmap related to NR-AGC is shown in Fig. S4.

To investigate the interpretability and classification basis of the federated learning radiomic features for the two categories, a category visualisation was conducted. Figure 4a shows the federated feature visualisation images of eight patients, including both nonrecurrent and recurrent patients. The federated learning radiomic features exhibited a greater lesion focus in the recurrence category than in the no-recurrence category. The Euclidean distance between the common and adaptive features across the four centres demonstrated that the similarity among common features was greater than that among adaptive features (Fig. 4b). Figure 4c shows the prediction scores of the RFLM for the two categories, which demonstrated a noticeable difference in prediction scores, enabling effective differentiation between the two subtypes (R-AGC and NR-AGC).

### RFLM ablation experiment

To verify the effectiveness of each component in the RFLM, ablation studies were performed on the components of the RFLM, except for the GAN component, which relies on the GCN component to strengthen the algorithm. The specific AUC results are shown in Table 2.

In the ablation experiments, Groups 2 and 4 showed that the spatial attention mechanism introduced by CBAM in the Resnet18 network better extracted spatial information from lesions, thereby improving the diagnostic performance of the model. Groups 3 and 4 indicated that using GCN networks to learn domain-specific information from different datasets and incorporating this domain-specific information into robust strategies effectively improved the diagnostic performance across all four centres. In the ablation experiments, when GAN components were not used, the Euclidean distances between the parameters of each central model were used as the robust input matrix.

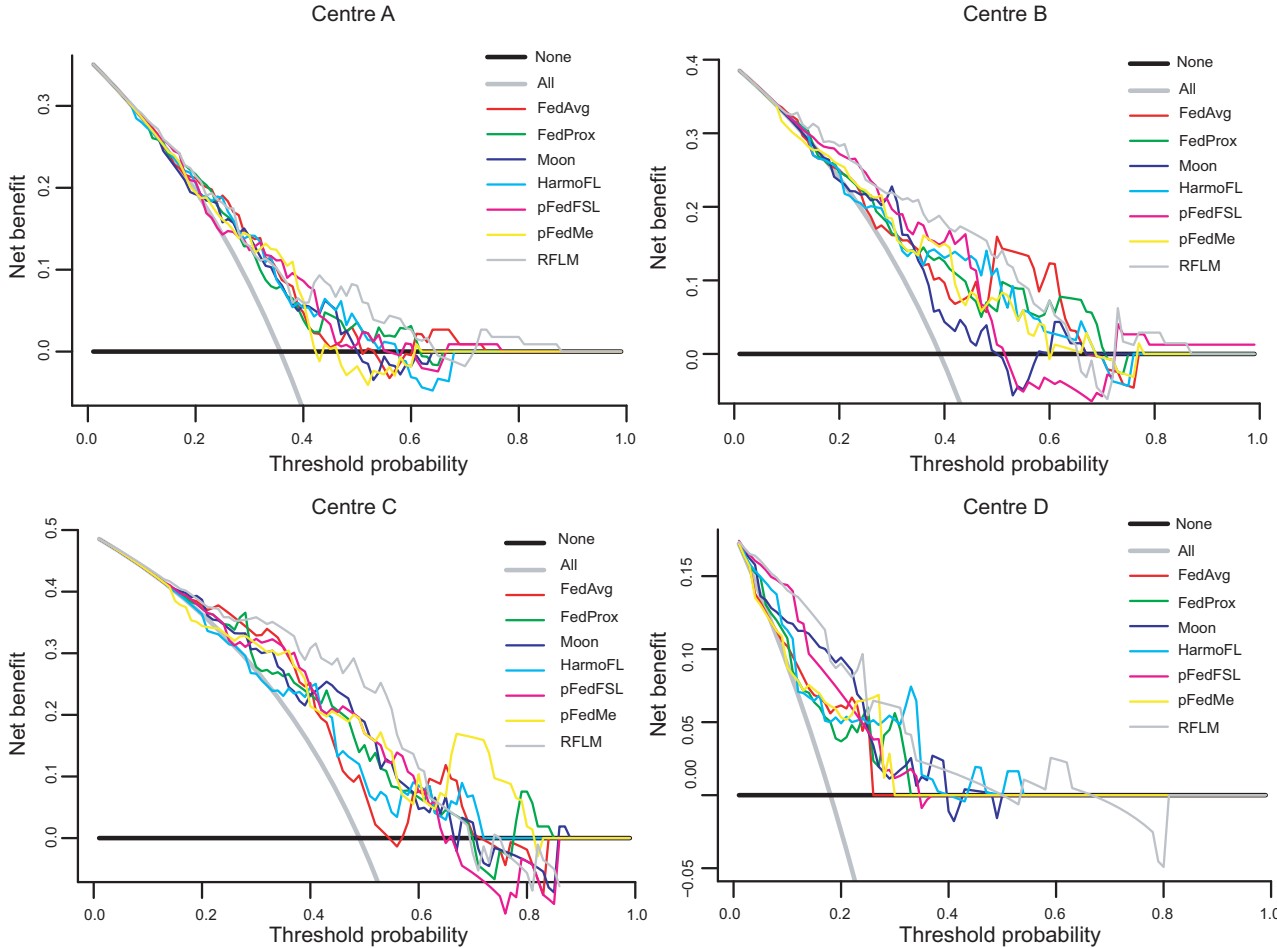

**Fig. 1 | DCA curves of seven models using data from four data centres.** The solid grey line assumes that all patients were involved in the R-AGC group while the black line assumes no patients were involved. The threshold probability was the point where the expected benefit of the treatment and treatment avoidance were equal.

The result showed that the net benefit of the RFLM was greater than that of the clinical model (range, 0.00–1.00). RFLM robust federated learning model, FedAvg, Fedprox, Moon, HarmoFL, pFedFSL and pFedMe are the comparison test algorithms.

## Research on other tasks

To demonstrate the applicability of the proposed RFLM algorithm to other tasks, a dataset collected from the Lung Image Database Consortium (LIDC-IDRI) for lung cancer diagnosis (https://wiki.cancerimagingarchive.net/display/Public/LIDC-IDRI) was utilised. The LIDC data was collected from 7 research institutions and 8 medical imaging companies, resulting in a total of 1018 patients. The malignancy of lung nodules ≥3 mm can be classified into grades 1–5, with grade 3 being an indeterminate malignancy[20]. In the present study, lesions with a malignancy of Grades 1–2 were classified as benign, while those with a malignancy of Grades 4–5 were classified as malignant. In total, 1746 lesions were included in the dataset. Due to the removal of medical information from each centre in the LIDC dataset, the entire LIDC dataset was divided into four groups, labelled A to D, using a random grouping approach to evaluate the performance of the RFLM algorithm. The LIDC data distribution is shown in Table S4.

The same preprocessing procedures were then applied to both the LIDC and multicentre gastric cancer datasets, and the results were validated using the RFLM algorithm. The diagnostic performance is shown in Table 3.

## Discussion

Accurate prognosis assessment plays a crucial role in determining individualised and precise treatments for gastric cancer patients. Currently, the TNM staging system[21,22] is commonly used for prognosis evaluation in gastric cancer patients. This system categorises patients based on the depth of tumour invasion, extent of lymph node involvement, and presence of metastasis, providing an estimation of the patient's risk level. However, relying solely on the TNM staging system for prognosis assessment may not provide a comprehensive evaluation of the risk of recurrence. The TNM staging system primarily focuses on tumour anatomical characteristics and lacks consideration of tumour heterogeneity and patient-specific predictive information. Consequently, achieving individualised and precise assessment becomes challenging using this system alone. To improve the accuracy of prognosis assessment, additional factors that account for tumour heterogeneity and patient-specific predictive information must be incorporated.

In recent years, computer-aided diagnosis has played a significant role in studying the postoperative recurrence of gastric cancer. Zhou et al. utilised machine learning methods and established models, such as random forest, decision tree, and logistic regression models, to predict postoperative recurrence of gastric cancer; they combined factors, such as BMI, operation time, weight, and age, to achieve AUC values of 0.922 in the training cohort (with 1607 patients) and 0.771 in the test cohort (with 405 patients)[23]. Jiang et al. developed a deep neural network, called S-net, for predicting disease-free survival in gastric cancer patients; they evaluated a cohort of 457 patients and achieved AUC values ranging from 0.792 to 0.802, and they also conducted a larger cohort study, including 1615 patients[24].

The previous methods were based on single-centre data, in which fixed machines and image acquisition protocols were used. However,

**Table 1 | AUC of the seven models using four central datasets of patients in the relapse and nonrelapse models**

| Method | Evaluation | Centre A | Centre B | Centre C | Centre D |
|--------|-----------|----------|----------|----------|----------|
| FedAvg | AUC | 0.672 | 0.726 | 0.711 | 0.798 |
| | Accuracy | 0.634 | 0.536 | 0.660 | 0.639 |
| FedProx | AUC | 0.658 | 0.718 | 0.731 | 0.766 |
| | Accuracy | 0.607 | 0.681 | 0.660 | 0.557 |
| Moon | AUC | 0.663 | 0.661 | 0.724 | 0.775 |
| | Accuracy | 0.634 | 0.565 | 0.679 | 0.689 |
| HarmoFL | AUC | 0.684 | 0.723 | 0.707 | 0.773 |
| | Accuracy | 0.616 | 0.696 | 0.604 | 0.820 |
| pFedMe | AUC | 0.689 | 0.706 | 0.744 | 0.769 |
| | Accuracy | 0.634 | 0.681 | 0.623 | 0.787 |
| pFedFSL | AUC | 0.649 | 0.742 | 0.728 | 0.796 |
| | Accuracy | 0.652 | 0.623 | 0.679 | 0.656 |
| RFLM | AUC | 0.710 | 0.798 | 0.809 | 0.869 |
| | Accuracy | 0.598 | 0.710 | 0.755 | 0.689 |

*RFLM* robust federated learning model.

in scenarios involving multiple centres, institutions, and operators, traditional deep learning methods and clinical diagnostic models often face challenges due to data heterogeneity, resulting in a decline in diagnostic performance. Several issues contribute to this situation. First, diagnostic models based on clinical experience are susceptible to significant interobserver variability, leading to poor diagnostic consistency among multiple physicians. This inconsistency can impact the overall diagnostic performance when applied across different centres. Second, the limited number of patients whose clinical information was collected from a single centre may not accurately capture the decisive role of certain clinical indicators in postoperative gastric cancer recurrence. This limitation may hinder the generalisability of the diagnostic models to different populations. Finally, different centres may employ distinct data collection devices and may be subject to geographical variations. As a result, the data collected may exhibit nonindependent and non-IID features. The inconsistency in the data distributions between different images poses challenges for model development and generalisation.

In the present study, the developed RFLM demonstrated superior performance compared to both the clinical model and other federated learning algorithms, demonstrating its effectiveness in predicting

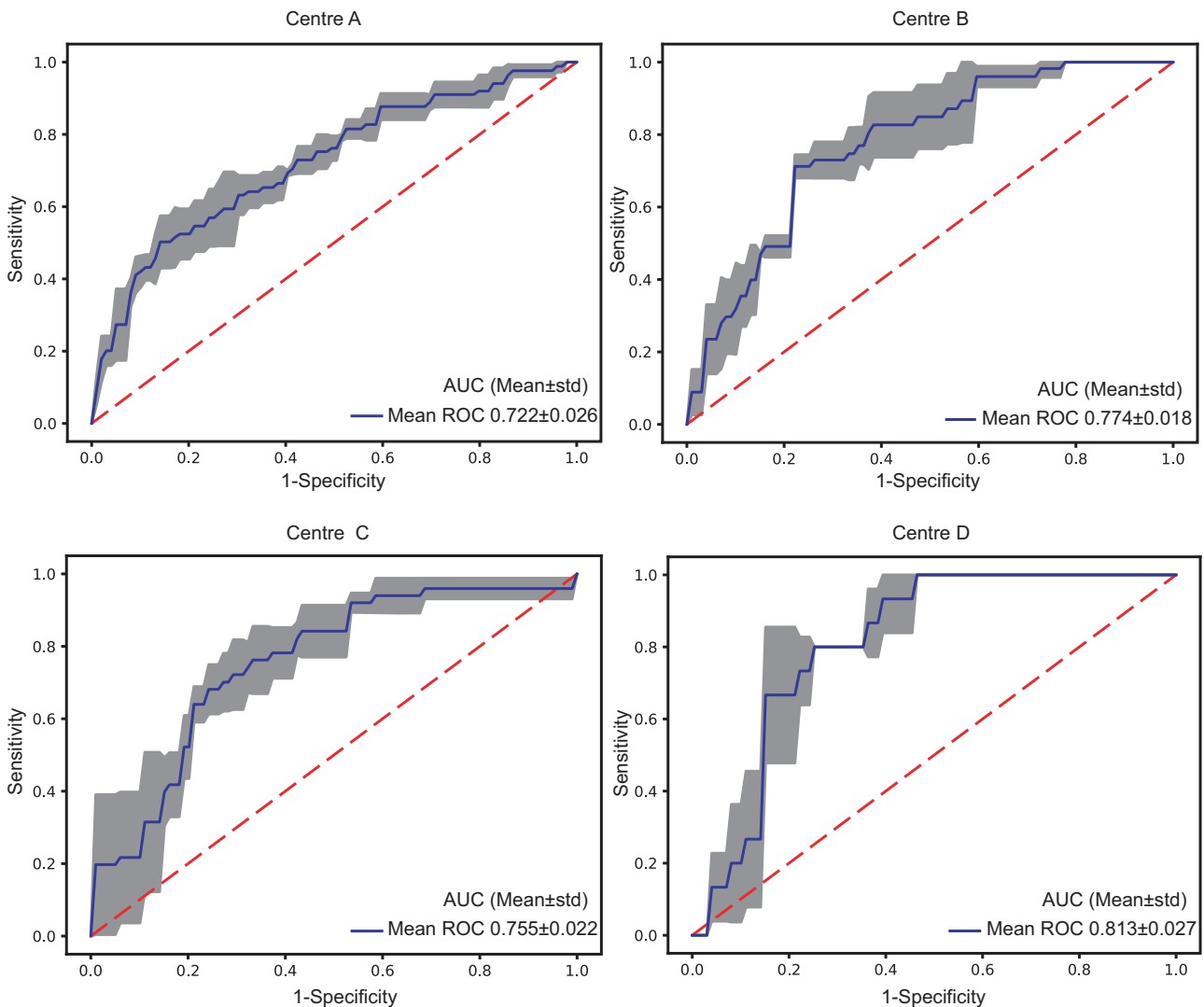

**Fig. 2 | Threefold cross-validation ROC curves for the four centres.** The blue curve represents the average AUC of the threefold curve. The grey areas represent the upper and lower limits of the ROC curve. The error band in the grey areas is the upper and lower boundary of the three-fold cross-verified ROC curve. Mean the AUC average for three-fold cross-validation, Std standard deviation.

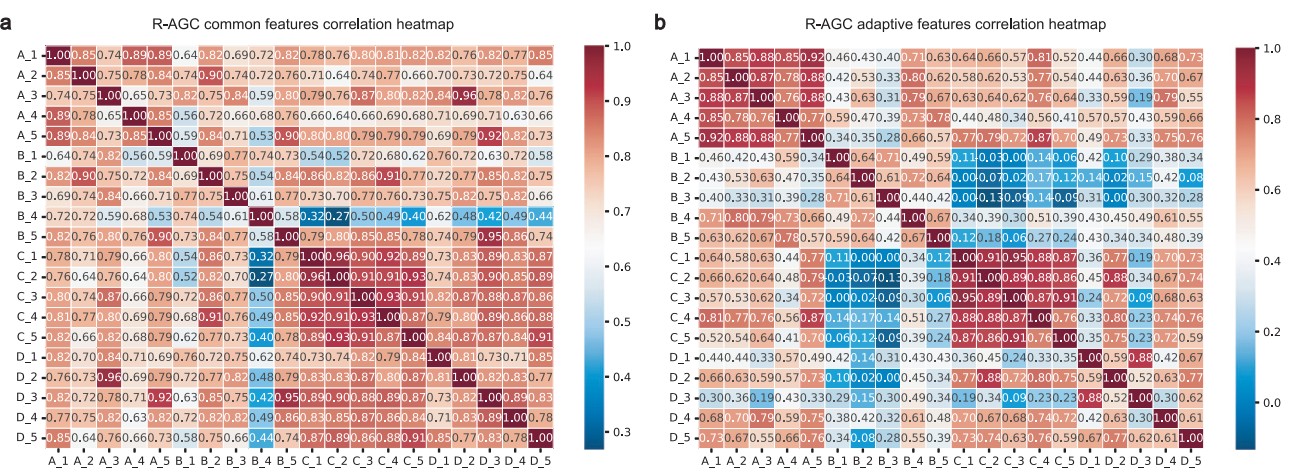

**Fig. 3 | Correlation heatmap of common recurrence features and adaptive features. a** R-AGC common features correlation heatmap. **b** R-AGC adaptive features correlation heatmap. A_1 first feature at centre A, R-AGC recurrent advanced gastric cancer.

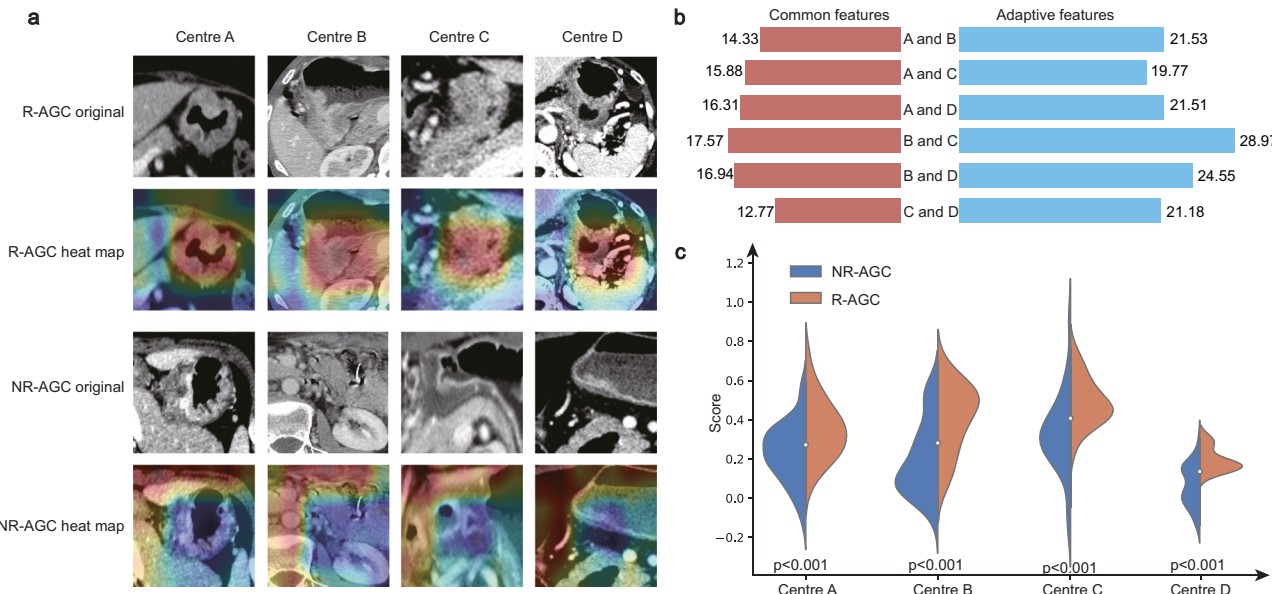

**Fig. 4 | RFLM Algorithm Result Analysis Diagram. a** The heatmap shows the information acquired by the RFLM for images in the recurrent and nonrecurrent classes. The red areas indicate a high level of model attention, while the blue areas indicate a low level of model attention. **b** The Euclidean distance plots depict the distance between the common and adaptive features of the four central data points. The left side represents common features, while the right side represents adaptive features. **c** The score charts illustrate the positive and negative images of the four data centres evaluated by the RFLM. Statistical test: Independent *t*-test (two-tailed). RFLM robust federated learning model, NR-AGC no recurrent advanced gastric cancer, R-AGC recurrent advanced gastric cancer, *p* significance value.

postoperative recurrence in gastric cancer patients. Additionally, compared to all treatment or nontreatment options, the RFLM provided better outcomes for patients after gastric cancer surgery. Robustness testing further validated the performance of the RFLM, as it exhibited good resistance to interference. The following AUC value ranges were observed after robustness testing: 0.700–0.717, 0.750–0.799, 0.750–0.809, and 0.826–0.846.

The proposed offers several advantages. First, the RFLM ensures the privacy and security of patient data, which is achieved by utilising a Wasserstein generative adversarial network (WGAN) to generate a partial representative dataset during the robust process. This approach allows the introduction of interdomain information and topological structure information from data from different centres, enhancing the generalisation performance of the model across multiple central institutions. Figure 5 shows a portion of the representative

dataset generated using the WGAN. Second, the RFLM leverages domain-related information and the topological relationships among data during the robust training of different local models through graph convolutional networks (GCNs). The local models within the RFLM incorporate domain-specific information and capture the topological structure from the data of different centres, enabling them to effectively capture adaptive features from local data.

In addition to its advantages, the RFLM also demonstrated strong diagnostic performance and robustness. The RFLM effectively identifies the most important common features that exhibit a high degree of similarity between each data centre, allowing for accurate differentiation between recurrent and nonrecurrent stomach cancer across different data centres. Simultaneously, the RFLM identifies adaptive features that exhibit a high degree of similarity within each data centre and a low degree of similarity within each data centre, facilitating local

data differentiation. Compared to other robust federated learning algorithms, the RFLM ensures data privacy by preserving the privacy of multicentre data by incorporating information from other data centres during the robust process, not sharing the original CT images between data centres, and relying solely on model parameters, including those trained on CT images (generated through the WGAN). This approach minimises the disparity between the global model and the robust model to the fullest extent. By mitigating the interference caused by heterogeneous features on local models, the RFLM effectively addresses the issue of parameter drift between the global centre and local models. Moreover, previous studies have reported a postoperative recurrence rate for AGC ranging from 20.1 to 50.7%[25,26]. This variance results in a substantial disparity in sample proportions between data centres for the two types (NR-AGC and R-AGC), potentially introducing bias into the model outcomes. To mitigate the impact of imbalanced data samples, the present study employed focal loss during model training to mitigate the bias towards a larger sample, thereby alleviating the impact of imbalanced data samples.

The present study had several limitations. First, the present did not explore intercentre similarities, indicating that further investigations are needed to explore potential similarities among datasets from different centres to enhance the robustness of the proposed approach by revealing common features and identifying areas where the model can effectively leverage data from multiple centres. Second, the present study did not characterise the common and adaptive features. It is important to feature the knowledge incorporated in robust models from different centres. Specifically, understanding whether common features originate from all centres collectively while adaptive features are solely derived from local data would provide valuable insights into the mechanisms underlying the robust federated learning framework.

## Methods
### Patients
The present study adopted a retrospective data collection methodology and included patients diagnosed with gastric cancer, which was confirmed by surgical pathology, from April 2008 to November 2019

### Table 2 | AUC results for the RFLM ablation experiment

| CBAM | FED | GCN | GAN | Cohort | Centre A | Centre B | Centre C | Centre D |
|---|---|---|---|---|---|---|---|---|
|  | √ | √ |  | Train | 0.745 | 0.847 | 0.803 | 0.853 |
|  |  |  |  | Test | 0.650 | 0.787 | 0.708 | 0.782 |
|  | √ | √ | √ | Train | 0.719 | 0.783 | 0.862 | 0.941 |
|  |  |  |  | Test | 0.654 | 0.709 | 0.724 | 0.807 |
| √ | √ | √ |  | Train | 0.751 | 0.778 | 0.805 | 0.802 |
|  |  |  |  | Test | 0.688 | 0.685 | 0.719 | 0.762 |
| √ | √ | √ | √ | Train | 0.750 | 0.814 | 0.811 | 0.875 |
|  |  |  |  | Test | 0.710 | 0.798 | 0.809 | 0.869 |

*RFLM* robust federated learning model, *FED* conventional federated learning, *GCN* graph convolutional neural network, *GAN* generative adversarial network.

### Table 3 | LIDC multicentre diagnostic performance

| Method | Evaluation | Centre A | Centre B | Centre C | Centre D |
|---|---|---|---|---|---|
| RFLM (Train) | AUC | 0.865 | 0.842 | 0.868 | 0.831 |
|  | Sensitivity | 0.732 (52/71) | 0.761 (67/88) | 0.781 (64/82) | 0.776 (52/67) |
|  | Specificity | 0.907 (166/183) | 0.833 (145/174) | 0.806 (179/222) | 0.778 (140/180) |
|  | Accuracy | 0.858 (218/254) | 0.809 (212/262) | 0.799 (243/304) | 0.777 (192/247) |
|  | PPV | 0.754 (52/69) | 0.698 (67/96) | 0.598 (64/107) | 0.565 (52/92) |
|  | NPV | 0.897 (166/185) | 0.874 (145/166) | 0.909 (179/197) | 0.903 (140/155) |
| RFLM (Test) | AUC | 0.816 | 0.811 | 0.852 | 0.824 |
|  | Sensitivity | 0.681 (32/47) | 0.650 (26/40) | 0.807 (25/31) | 0.778 (28/36) |
|  | Specificity | 0.803 (98/122) | 0.812 (121/149) | 0.763 (71/93) | 0.684 (108/158) |
|  | Accuracy | 0.796 (130/169) | 0.778 (147/189) | 0.774 (96/124) | 0.701 (136/194) |
|  | PPV | 0.571 (32/56) | 0.482 (26/54) | 0.532 (25/47) | 0.359 (28/78) |
|  | NPV | 0.867 (98/113) | 0.896 (121/135) | 0.922 (71/77) | 0.931 (108/116) |

*AUC* area under the curve, *PPV* positive predictive value, *NPV* negative predictive value.

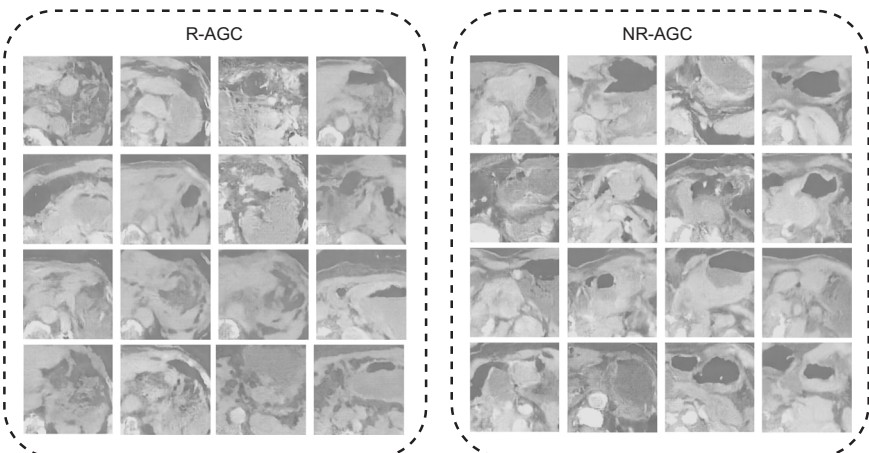

**Fig. 5 | A representative dataset was generated based on the WGAN.** NR-AGC nonrecurrent advanced gastric cancer, R-AGC recurrent advanced gastric cancer.

**Table 4 | Basic patient information**

| Centre | Set | Disease type | Gender | | Age (Mean ± Std) | N-stage | | | | | T-stage | | | | CA199 | |
|---|---|---|---|---|---|---|---|---|---|---|---|---|---|---|---|---|
| | | | Male | Female | | 0 | 1 | 2 | 3a | 3b | 1 | 2 | 3 | 4 | Absent | Present |
| Centre A (293) | Train (181) | R-AGC (51) | 33 | 18 | 59.84 ± 11.80 | 8 | 4 | 15 | 10 | 14 | 0 | 2 | 32 | 17 | 45 | 6 |
| | | NR-AGC (130) | 88 | 42 | 60.28 ± 12.85 | 39 | 24 | 20 | 40 | 7 | 0 | 13 | 79 | 38 | 112 | 18 |
| | Test (112) | R-AGC (40) | 25 | 15 | 60.73 ± 10.36 | 4 | 1 | 15 | 13 | 7 | 0 | 0 | 25 | 15 | 34 | 6 |
| | | NR-AGC (72) | 39 | 33 | 60.19 ± 12.59 | 22 | 11 | 12 | 18 | 9 | 0 | 8 | 32 | 32 | 61 | 11 |
| Centre B (140) | Train (71) | R-AGC (22) | 17 | 5 | 65.63 ± 10.28 | 3 | 3 | 6 | 7 | 3 | 0 | 0 | 13 | 9 | 17 | 5 |
| | | NR-AGC (49) | 33 | 16 | 61.27 ± 11.20 | 19 | 9 | 9 | 8 | 4 | 0 | 6 | 23 | 20 | 44 | 5 |
| | Test (69) | R-AGC (27) | 22 | 5 | 63.27 ± 10.35 | 4 | 2 | 7 | 7 | 7 | 0 | 2 | 17 | 8 | 25 | 2 |
| | | NR-AGC (42) | 25 | 17 | 61.35 ± 11.02 | 7 | 10 | 12 | 6 | 7 | 0 | 8 | 21 | 13 | 34 | 8 |
| Centre C (109) | Train (56) | R-AGC (24) | 15 | 9 | 58.46 ± 12.74 | 7 | 3 | 5 | 6 | 3 | 0 | 6 | 5 | 13 | 21 | 3 |
| | | NR-AGC (32) | 19 | 13 | 54.84 ± 10.46 | 13 | 6 | 5 | 7 | 1 | 0 | 8 | 6 | 18 | 28 | 4 |
| | Test (53) | R-AGC (26) | 16 | 10 | 58.19 ± 13.64 | 7 | 4 | 7 | 6 | 2 | 0 | 7 | 6 | 13 | 17 | 9 |
| | | NR-AGC (27) | 14 | 13 | 53.26 ± 13.17 | 8 | 8 | 6 | 4 | 1 | 0 | 8 | 3 | 16 | 22 | 5 |
| Centre D (99) | Train (38) | R-AGC (4) | 2 | 2 | 53.75 ± 12.45 | 2 | 1 | 0 | 1 | 0 | 0 | 2 | 1 | 1 | 3 | 1 |
| | | NR-AGC (34) | 20 | 14 | 57.85 ± 11.66 | 11 | 8 | 6 | 5 | 4 | 4 | 7 | 9 | 14 | 30 | 4 |
| | Test (61) | R-AGC (11) | 7 | 4 | 54.64 ± 13.68 | 1 | 0 | 5 | 5 | 0 | 1 | 0 | 8 | 2 | 8 | 3 |
| | | NR-AGC (50) | 26 | 24 | 55.00 ± 12.81 | 22 | 12 | 6 | 6 | 4 | 14 | 8 | 17 | 11 | 45 | 5 |

*NR-AGC* no recurrent advanced gastric cancer, *R-AGC* recurrent advanced gastric cancer, *Std* standard deviation.

across four medical centres. This study was implemented under the approval of the Jiangmen Central Hospital, Meizhou People's Hospital, The First Affiliated Hospital of Sun Yat-sen University, and Dongguan People's Hospital, and conducted in accordance with the 1964 Helsinki Declaration and its later amendments or comparable ethical standards. Informed consent was waived by our Institutional Review Board because of the retrospective nature of our study.

The inclusion criteria were as follows: (1) complete preoperative contrast-enhanced abdominal CT images were available; (2) the interval between the preoperative CT examination and surgery was less than 2 weeks; (3) gastric adenocarcinoma was confirmed via surgical pathology; (4) comprehensive clinical data and regular follow-up information were available; and (5) patients who experienced recurrence, with a minimum follow-up time of 2 years, were excluded.

The exclusion criteria were as follows: (1) poor-quality CT images where lesion visualisation was unclear; and (2) other malignant tumours detected via CT scans. The primary endpoints were local recurrence and nonrecurrence (NR) cases, with a minimum follow-up time of 5 years for all the other cases. Patients underwent follow-up appointments every 3–6 months during the first 2 years, every 6–12 months over the following 3 years, and annually thereafter. The primary follow-up methods included contrast-enhanced abdominal CT, gastroscopy, and tumour biomarker examinations. Following the aforementioned screening, a total of 641 patients were included in the study. The dataset was divided by a random method, and the basic patient information is listed in Table 4.

**Definition of gastric cancer recurrence**
According to the National Comprehensive Cancer Network (NCCN) Guidelines for Gastric Cancer (version 2.2019), the recurrence patterns of gastric cancer (GC) include locoregional recurrence (LR) and metastatic disease. Metastatic disease can be divided into peritoneal dissemination and distant metastasis. In the present study, patients were classified according to their recurrence pattern as LR, peritoneal metastasis, or distant metastasis. LR included recurrence in the gastric bed, gastric remnant of the anastomosis, duodenal stump, and/or lymph node recurrence in the gastric region. Peritoneal metastasis included metastasis in the peritoneum, omentum, and mesentery.

Distant metastases included those that occurred in other organs and nongastric regional lymph nodes[27,28].

Recurrence in the gastric bed and in the gastric remnant of the anastomosis was confirmed by gastroscopic biopsy. The recurrence of gastric region lymph nodes and duodenal stumps was mainly determined by dynamic follow-up observation via postoperative enhanced CT. During the follow-up with enhanced CT, when the lymph nodes in the gastric region were enlarged with necrosis or gradually enlarged during the dynamic follow-up, regional lymph node recurrence was considered after excluding tuberculosis and other factors. Peritoneal metastasis was considered to have occurred in the following instances: when postoperative CT examination revealed nodular or mass-like thickening of the peritoneum, omentum, or mesentery; when there was an increase in the number of foci or enlargement of the foci on dynamic follow-up; or when the ascites was positive for cancer cells. Distant metastasis was confirmed by dynamic postoperative CT observation.

According to the National Comprehensive Cancer Network (NCCN) Guidelines for Gastric Cancer (version 2.2019), most postoperative recurrences of GC occur within 2 years after surgery. Patients, except for recurrent patients, were followed up for at least 2 years in the present study. As shown in Fig. S5, the patients were followed up every 3–6 months in the first 2 years, once every 6–12 months in the following 3 years, and once a year thereafter. The main follow-up examinations included abdominal contrast-enhanced CT, gastroscopy, and tumour biomarker detection.

**ROI acquisition**
In the initial stage of the study, radiologists with more than 10 years of abdominal imaging diagnostic experience defined the region of interest (ROI) as the input for the client models. This process involved accurately outlining the contours of the lesion. Subsequently, a rectangular frame was constructed based on the exact contour of the lesion, encompassing the entire lesion boundary, making the selection of the ROI less susceptible to the subjective experiences of clinicians[29,30]. Deep learning methods can automatically extract ROIs, eliminating the need for precise manual delineation of ROIs. The details of the data preprocessing procedure are provided in Supplementary S1.

## Construction of the RFLM

Directly using the global model parameters for each local model may lead to poor parameter consistency among local models when addressing the nonindependent and independently distributed (non-IID) issues present in the data from different centres[31,32]. Therefore, in the search for common features, it is necessary to perform robust learning on the global model from the central model to adapt it to the local data of each centre. In the present study, the RFLM introduced two key improvements (Fig. 6).

On the premise of ensuring data security, a representative dataset reflecting the data characteristics of each centre was generated using the Wasserstein generative adversarial network (WGAN)[33]. This representative dataset was based on the data from each centre. Subsequently, a local model relationship matrix was created by each client's local model using the representative dataset, which provided the preference level of each local model for the same data and reflected the correlation information between the domains of each local model. Furthermore, the present study considered the potential risk of privacy leakage from the uploaded data generated by the WGAN. To address this issue, a random Gaussian perturbation matrix was added to the data generated by the WGAN, thus ensuring the privacy protection of the local data. Detailed information for the WGAN is provided in Supplementary S2.

Based on the acquired local model relationship matrix, graph convolutional networks (GCNs)[34] were employed to generate an adjacency matrix. This adjacency matrix captured the related information among local models from each data centre. Subsequently, the adjacency matrix was combined with each local model to obtain robust local models. These improvements enhanced the generalisability of the model by incorporating client-specific information from different clients' data, which allowed the model to learn from the unique characteristics and patterns present in each client's data, leading to more robust and effective predictions. Of note, all the client models utilised deep learning techniques in the present study. More detailed information for the algorithm is provided in Supplementary S3.

To enhance utilisation of the robust model features, each data centre utilised the convolutional kernels of its own robust local model as feature extractors. Multiple feature maps were extracted from the local CT image data of each patient within the local dataset. Subsequently, the average value of each feature map was computed to create a unified radiomic feature (Fig. S6). Because the robust local model consisted of a total of 4449 convolutional kernels, a total of 4449 federated radiomic features were extracted, serving as the basis for the RFLM classification task. To select the most significant and discriminative radiomic features, the Mann–Whitney $U$-test was applied, which identified features that exhibited significant differences between the two groups. Subsequently, the maximum relevance minimum redundancy (mRMR) algorithm was used to further filter and retain the most valuable radiomic features. Finally, the constructed RFLM utilised a sparse Bayesian extreme learning machine[35] (Fig. S7). Due to the positive and negative samples, the results of the overall model may be biased, which will affect the overall performance of the model. Therefore, the focal loss[36] was used for all loss functions in the present study to alleviate the problem of a large gap between the two types of samples.

## Assessment of common and adaptive features of the RFLM

To explore the basis of inference from the robust models of each centre and identify common features for class predictions, as well as adaptive features for the differences between centre data, the present study extracted federated radiomic features from each centre using robust models. Specifically, four sets of federated radiomic features were extracted from all the data samples using individualised models.

The Mann–Whitney $U$-test and the mRMR algorithm were subsequently applied to select the 200 most valuable radiomic features from each feature set. The correlations between the four sets of federated radiomic features were then computed. The features displaying the highest correlation within each data centre and between different data centres were identified as adaptive and common features, respectively. The present study used the Pearson correlation coefficient[37], and detailed information is provided in Supplementary S4.

## Evaluation and comparison of models

To comprehensively evaluate the performance of the RFLM in a multicentre setting, the present study conducted a comparison with other models, including a clinical feature-based CM model, traditional federated learning models (FedAvg[15], FedProx[16], Moon[17], and HarmoFL[18]), and the latest robust federated learning models (pFedMe[14] and pFedFSL[19]). In the present study, the clinical features included T stage, N stage, and CA199 status, and the random forest algorithm was used to construct clinical models for the four data centres (Supplementary S5). The details of the subjective CT and pathological assessments are provided in Supplementary S6.

To further validate the robustness of the RFLM, five rounds of data shuffling and threefold cross-validation across multiple centres were performed. This process involved random permutations of the dataset distribution among the centres to comprehensively assess the impact of changes in the data distribution on model performance.

## Common and adaptive features in federated radiomics

For the evaluation of common and adaptive features in federated radiomics, categorical heatmaps were utilised to construct visual representations of the attention given by the RFLM to the two types of image data from different centres. The heatmaps provided insights into how the model focused on common and adaptive features within the data. In addition, the similarity between the adaptive and common features was assessed by calculating the correlation matrix of federated radiomic features across different centres, providing an understanding of the relationships and similarities among the features across centres. Moreover, the Euclidean distance between features was computed to quantify the dissimilarity between adaptive and common features, which provided valuable information about the differences and variations present among the features. By employing these methods, the present study evaluated the common and adaptive features in federated radiomics, shedding light on the attention and relationships of the RFLM towards different types of features and data from various centres.

## Statistical analyses

To comprehensively evaluate the performance of various algorithms, quantitative indicators, such as the area under the curve (AUC), specificity, sensitivity, accuracy (ACC), positive predictive value (PPV), and negative predictive value (NPV), were utilised to validate the prediction results. The integrated discrimination improvement (IDI) and net reclassification index (NRI) were also used to assess whether the RFLM exhibited statistically significant differences in predicting postoperative gastric cancer recurrence. These evaluation metrics provided a comprehensive assessment of the predictive capabilities of the algorithms and allowed for statistical comparisons between different models. The ROC curve was used to illustrate the overall performance of the different modelling methods, and DCA was used to evaluate the clinical effectiveness of the model in predicting postoperative recurrence of gastric cancer. The robustness test suggested that the model was less affected by the local data distribution.

Statistical analyses were conducted using two-tailed tests, and a $p$ value < 0.05 was considered statistically significant.

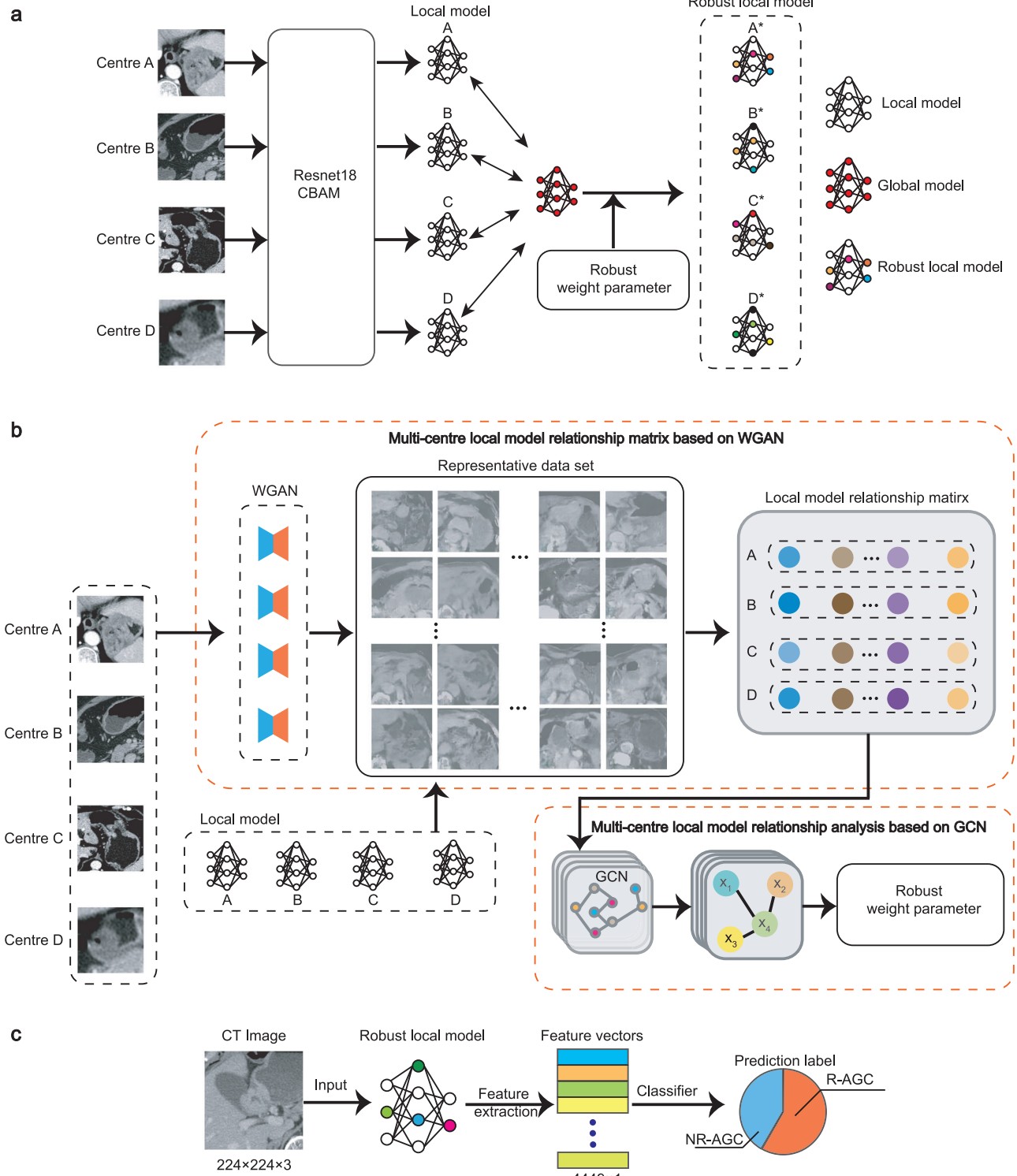

**Fig. 6 | RFLM algorithm diagram. a** Construction process of the robust local model in the RFLM. **b** Details of generating robust weight parameters in the robust local model. **c** Feature extraction and feature classification in the RFLM. CBAM convolutional block attention module, NR-AGC nonrecurrent advanced gastric cancer, R-AGC recurrent advanced gastric cancer, GCN graph convolutional neural network, WGAN Wasserstein generative adversarial network.

## Experimental equipment

Statistical analysis was performed using R software (version 3.4.0, http://www.Rproject.org) and IBM SPSS Statistics 20.0 software (SPSS, Chicago, IL, USA).

For deep learning tasks, an NVIDIA RTX A6000 graphics card with CUDA version 10.2 and 48 GB of GPU memory was utilised. The deep learning framework used was PyTorch 1.7.1 with graphics processing unit (GPU) support, implemented in Python 3.6. Additionally, MATLAB version 2020b was used for certain analysis tasks.

## Reporting summary

Further information on research design is available in the Nature Portfolio Reporting Summary linked to this article.

## Data availability

The Multicentre CT data of gastric cancer in the current study are not publicly available for patient privacy policy. However, if researchers wish to access our data solely for scientific research purposes, the corresponding author can share the relevant data. Requests will be processed by the corresponding author within 3 months and followed up with the requesting party. Any requests will be pending prior approval and revision by the Ethics Committee of Jiangmen Central Hospital, the Ethics Committee of Meizhou People's Hospital, the Ethics Committee of The First Affiliated Hospital of Sun Yat-sen University, and the Ethics Committee of Dongguan People's Hospital, which retain all rights to deny access. Additionally, we have used publicly available LIDC dataset in this study as Supplementary Data 1. The deidentified data generated during and/or analysed during the current study are provided as source data file. Source data are provided with this paper.

## Code availability

The codes are provided at GitHub (https://github.com/baofengguat/RFLM-project/tree/master). Supplementary Code is a detailed supplement to the article code.

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

## Acknowledgements

This work was supported by the National Natural Science Foundation of China (81960324, 62176104, 12261027), the Natural Science Foundation of Guangxi Province (2021GXNSFAA075037), the incubation project of 1000 Young and Middle-aged Key Teachers in Guangxi Universities (2018GXOGFB160).

## Author contributions

B.F. conceived and designed the study, developed all methods, and drafted the manuscript. J.S. implemented the methodology, conducted experiments, and contributed to manuscript writing. L.H. participated in relevant medical research, and contributed to manuscript writing. Z.Y., S.-T.F., J.L., Q.C., H.X., X.C. and C.W. collected multicenter experimental data and information. Q.H. provided assistance for engineering experiments. E.C. offered theoretical support for medical research. Y.C. contributed to the design of the methodology and manuscript writing. W.L. provided overall support for the research and contributed to method design. All authors reviewed and revised the manuscript.

## Competing interests

The authors declare no competing interests.
