## [Peer Review File · Nature Communications]

Robustly Federated Learning Model for Identifying High-Risk Patients with Postoperative Gastric Cancer RecurrenceREVIEWER COMMENTS

Reviewer #1 (Remarks to the Author):

This study explored establishing a robust federated learning model to predict postoperative gastric cancer recurrence based on CT imaging data in a four centers dataset. The authors articulate several advantages of their approach, including the handling of multi-center data, privacy preservation through federated learning, leveraging common and unique knowledge, and interpretability through visualizations. These merits are commendable and demonstrate the potential significance of the research in the field of medical image analysis and cancer recurrence prediction.

But I have concerns about this study:

Main issues:

1. How did they define and evaluate the recurrence after surgery? especially concerning specific types of recurrence (e.g., peritoneal recurrence) and their distribution of different recurrence types in the dataset. Additionally, details regarding the duration and frequency of follow-up visits for recurrence assessment should be provided.
2. They need to provide a table for sufficient clinical information for all patients, such as tumor stage, gender, and age distributions, which are essential for understanding patient characteristics and potential biases.
3. Limited exploration of inter-center similarities and knowledge characterization.
4. Small sample size from some centers limits robustness assessment. The authors should provide more discussions about this issue to clarify the potential concerns.
5. The class imbalance between centers may affect federated learning convergence. This issue is very important for clinical application, but this work did not provide comprehensive evaluations or discussions.
6. If possible, the authors can provide more ablation studies to analyze the contribution of each sub-component in the proposed method.
7. How many radiologists defined the region of interest (ROI)? For gastric cancer, it is not easy to accurately identify the tumor area for some cases, how did they deal with that?
8. 4,449 federated radiomics features are extracted in the paper, what are these features?

Reviewer #2 (Remarks to the Author):

This manuscript shows impressive results on a cross-silo federated learning problem of predicting gastric cancer recurrence. The results are promising, but I believe the manuscript needs to be improved and clarified in several respects.

The organization of the paper makes it somewhat difficult to read. For example, there is much discussion of "features" before the details of the model have been introduced to actually define the features. Too many different adjectives are used to describe features (e.g., "common features", "distinct features", "individual features", "common knowledge/information", "personality information", "federated group learning features", "robust features"). These need to be consolidated and precisely defined before being discussed. Similarly, the discussion in the "Result" section is difficult to read before the details of the machine learning model and setting are provided.

More details are also needed in order to be able to evaluate the privacy properties of the proposed approach. Following [1, Sec 1.2], it is recommended to include pseudo-code that makes it clear exactly what information is exchanged between the centers and the central server. In particular, it has been well-established that deep networks can memorize their training data (See for example [2], as well as many other works on private GANS), and so the use of the WGAN without additional techniques like differential privacy may not be sufficient to preserve privacy.

The relationship between RFLM and standard federated learning algorithms like FedAvg is not clear. FedAvg is not a specific ML model, but rather a model parameter optimization algorithm that can be applied to an arbitrary model architecture in the federated learning setting. However, RFLM appears to imply a particular set of model architectures (including feature extractors and a final classifier), as well as a particular recipe for training those models. If the RFLM algorithm's improvements over other FL methods are a major contribution of the paper that can be applied to other FL tasks, then detailed evaluation on public benchmark datasets should be included to support this point.

Tables 1-2 and Figures 1-2 seem to use a single fixed train/test split as given in Table 3.

However, given the relatively small number of examples, using 4 significant digits seems unjustified for these metrics. Further, rather than reporting the metrics for a single train/test split, it would be preferable to use cross-validation, and train multiple models on different train/test splits so the mean metrics could be reported together with appropriate confidence intervals. Fig 3 (b) appears to partially address this. However, the exact experiment is unclear; "five random permutations" in particular doesn't make sense to me, as a permutation refers to a re-ordering of the data, but Fig 3(a) clearly shows re-assignment of the examples to different train/test splits. However, in the 5 distributions, the ratio of train/test seems to vary significantly --- for cross validation, I would have expected a fixed train/test ratio, and just different partitions. I would suggest removing Fig 3 and simply using cross-validation to provide confidence intervals in Tables 1-2 and Figures 1-2, which would free up space to improve clarity on the other points I have raised.

It is unclear to me how the subjective CT findings (line 294) are used. Are these five CT signs (line 303) then used as features in the RFLM model, or does the model only use the ROI from the CT images as the raw input features for prediction? Or were these signals used in the clinical model?

Finally, the future of this work is unclear. While the results are promising, I would like to see a more robust discussion of what would be necessary to see these models actually deployed in a clinical setting. For example, do the models do well when applied to a center where they were not trained? This could be testing by a 4-way cross validating, training on 3 of the centers and evaluating on the final held-out center. Does the model still work if a different radiologist selects the ROIs for the training data vs the test data?

Comments on specific lines:

line 69: "without exposing data privacy" should perhaps be "exposing private data". But more importantly, the authors should clarify more explicitly exactly what the privacy benefit is. For example, can they make statements like "raw CT images are never shared between centers; instead model parameters (including a generative WGAN trained on the CT images)

are exchanged."

line 143: How is "correlation" between features computed? It would help to know the mathematical type of the features, and give a precise formula if possible.

line 168: Unclear what "attention" refers to

line 241: "inter-group / intra-group" -> it should be clarified exactly what "group" refers to. I believe this is either R or NR (the label / class in typical machine learning terminology), but that isn't completely clear, as it could be confused with site.

line 242: "two tasks" -> what are the tasks? does this mean the groups R and NR? Again, use consistent terminology and define it clearly early on, for example "we consider two groups (classes, labels) throughout, NR (no-recurrent) and R (recurrent)""

line 323: "client models" The paper has used "center" consistently to refer to the four medical centers. I believe "client" here is a synonym for "center". While "client" is often used in the FL literature, it would be more clear to pick one term and stick to it throughout the paper, remarking on the connection to other terminology when first introduced.

line 352: What is the type of these features? For example, is each feature value a single number? a vector or matrix of a specific size? To make this concrete, please clarify.

line 373: It's unclear how these 4 groups of 200 features relate to previous discussion for example in Fig. 4 and Fig. 5. What are Fig. 4's A_1 , ..., A_4 ?

References:

[1] Advances and Open Problems in Federated Learning.

<https://arxiv.org/pdf/1912.04977.pdf>.

[2] Extracting Training Data from Diffusion Models. <https://arxiv.org/pdf/2301.13188.pdf>.

Responses to Reviewer Comments

Reviewer #1: *This study explored establishing a robust federated learning model to predict postoperative gastric cancer recurrence based on CT imaging data in a four centers dataset. The authors articulate several advantages of their approach, including the handling of multi-center data, privacy preservation through federated learning, leveraging common and unique knowledge, and interpretability through visualizations. These merits are commendable and demonstrate the potential significance of the research in the field of medical image analysis and cancer recurrence prediction.*

Response:

First of all, we wish to thank you for the constructive, encouraging and positive comments. Your helpful suggestions have greatly improved our current work.

Question: *How did they define and evaluate the recurrence after surgery? especially concerning specific types of recurrence (e.g., peritoneal recurrence) and their distribution of different recurrence types in the dataset. Additionally, details regarding the duration and frequency of follow-up visits for recurrence assessment should be provided.*

Response:

Thank you very much for your valuable comments!

According to NCCN Guidelines for Gastric Cancer (Version 2.2019), the recurrence patterns of gastric cancer (GC) included locoregional recurrence (LR) and metastatic disease. The metastatic disease can be divided into peritoneal dissemination, and distant metastasis. Therefore, in this study, we classified recurrence pattern as LR, peritoneal metastasis and distant metastasis. LR includes recurrence in the gastric bed, gastric remnant of anastomosis, duodenal stump, and / or lymph node recurrence in the gastric region. Peritoneal metastasis includes metastasis in the peritoneum, omentum and mesentery. Distant metastasis included those occurred in other organs and non-gastric regional lymph nodes [1,2].

The recurrence in the gastric bed and gastric remnant of anastomosis was confirmed by gastroscopic biopsy. The recurrence of gastric region lymph node and duodenal stump was mainly determined by dynamic follow-up observation of postoperative enhanced CT. During the follow-up of enhanced CT, when the lymph nodes in the gastric region were enlarged with necrosis, or the lymph nodes in the gastric region were gradually enlarged during the dynamic follow-up, after excluding tuberculosis and other factors, the regional lymph node recurrence was considered. Peritoneal metastasis is considered to have occurred when postoperative CT examination reveals nodular or mass-

like thickening of the peritoneum, omentum, or mesentery, and when there is an increase in the number of foci or enlargement of the foci on dynamic follow-up, and/or when the ascites is positive for cancer cells. Distant metastasis was confirmed by postoperative CT dynamic observation.

According to NCCN Guidelines for Gastric Cancer (Version 2.2019), most postoperative recurrences of GC occur within two years after surgery. In this study, except for recurrent cases, the other cases were followed up for at least 2 years. As shown in **Figure A1**, the patients were followed up every 3 to 6 months in the first 2 years, once every 6 to 12 months in the following 3 years, and then once a year. The main follow-up examinations included abdominal contrast-enhanced CT, gastroscopy and tumor biomarkers.

In this study, a total of 91 cases of recurrence (91/293) occurred in center A, including 29 cases of LR, 17 cases of peritoneal metastasis, 53 cases of distant metastasis (including liver, lung, pancreas, gallbladder, kidney, bone, abdominal wall, and distant lymph nodes, etc.), and some of the cases combined more than two recurrence patterns. Center B had a total of 49 cases of recurrence (49/140), including 11 cases of LR, 12 cases of peritoneal metastasis, and 31 cases of distant metastasis (including liver, lung, bone, abdominal wall and distant lymph nodes, etc.), with some cases combining two or more recurrence patterns. Center C had 50 cases of recurrence (50/109), including 22 cases of LR, 9 cases of peritoneal metastasis, and 23 cases of distant metastasis (including liver, lungs, adrenal glands, and distant lymph nodes, etc.), with some cases combining two or more recurrence patterns. A total of 15 cases of recurrence occurred in center D (15/99), including 6 cases of LR, 5 cases of peritoneal metastasis, and 7 cases of distant metastasis (including liver, adrenal glands, abdominal wall and distant lymph nodes, etc.), with some cases combining two or more recurrence patterns.

(Manuscript: Definition of gastric cancer recurrence)

Figure A1. Disease free survival analysis of four data center test sets

Question: They need to provide a table for sufficient clinical information for all patients, such as tumor stage, gender, and age distributions, which are essential for understanding patient characteristics and potential biases.

Response:

Thanks for your thoughtful suggestion.

To assess the patient characteristics and potential biases more comprehensively, we provide the following clinical table for the four central datasets: (*Manuscript: Patient, Tabel 1: Basic information of patients*)

Tabel A1. Basic information of patients

Center	Set	Disease type	Gender		Age (Mean ±Std)	N-stage					T-stage				CA199	
			Male	Female		N0	N1	N2	N3a	N3b	T1	T2	T3	T4	Absent	Present
Center 1 (293)	Train (181)	Recurrence (51)	33	18	59.84±11.68	8	4	15	10	14	0	2	32	17	45	6
		Non-recurrence (130)	88	42	60.32±12.76	38	24	21	40	7	0	13	79	38	113	17
	Test (112)	Recurrence (40)	25	15	60.73±10.22	4	1	15	13	7	0	0	25	15	34	6
		Non-recurrence (72)	40	32	60.19±12.50	22	11	12	18	9	0	8	32	32	61	11
Center 2 (140)	Train (71)	Recurrence (22)	17	5	65.63±10.04	3	3	6	7	3	0	0	13	9	17	5
		Non-recurrence (49)	33	16	61.27±11.09	19	9	9	8	4	0	6	23	20	44	5
	Test (69)	Recurrence (27)	22	5	61.56±10.10	4	2	7	7	7	0	2	17	8	25	2
		Non-recurrence (42)	25	17	61.45±10.80	7	10	12	6	7	0	8	21	13	34	8
Center 3 (109)	Train (56)	Recurrence (24)	13	11	59.04±11.82	6	3	6	6	3	0	4	4	16	18	6
		Non-recurrence (32)	22	10	57.00±12.67	13	6	5	7	1	0	9	5	18	29	3

Center 4	Test (53)	Recurrence (26)	15	11	53.73±12.80	7	4	7	6	2	0	7	8	11	17	9
		Non-recurrence (27)	14	13	54.48±11.55	9	0	6	6	4	0	9	3	15	24	3
	Train (38)	Recurrence (4)	2	2	53.75±10.78	2	1	0	1	0	0	0	2	2	3	1
		Non-recurrence (34)	20	14	58.00±11.49	11	8	6	5	4	4	4	7	23	30	4
(99)	Test (61)	Recurrence (11)	7	4	55.00±13.04	1	0	5	5	0	1	1	8	2	8	3
		Non-recurrence (50)	26	24	55.00±12.68	22	12	6	6	4	14	8	17	11	45	5

Question: *Limited exploration of inter-center similarities and knowledge characterization.*

Response:

Thank you very much for your valuable comments!

We appreciate the reviewer's comment on the need for further exploration of inter-center similarities and knowledge characterization. In this study, we conducted the exploration of inter-center similarities and knowledge characterization from two aspects: model construction and feature analysis.

For inter-center similarities, as shown in Figure A2b, Euclidean distances [3] was introduced to measure the distance of samples between different centers. We calculate Euclidean distance common features and adaptive features between different centers, respectively.

For the feature analysis, the correlation heatmaps was constructed to reflect the relationships between centers' common features and adaptive features (as shown in **Figure A3**). To further explain the relationship between adaptive and common features across different data centers, we selected 5 adaptive features and 5 common features for each data center and constructed correlation heatmaps by calculating Pearson correlation coefficients. In these heatmaps, the red areas indicate stronger correlations, while the blue areas indicate weaker correlations. As shown in the heatmap of common feature correlations (a), there is a high degree of feature correlation across different data centers. In contrast, in the heatmap of adaptive feature correlations (b), the features within the same data center exhibit higher correlations. This further validates that adaptive features were more related the local data center, while common features are shared across different data centers.

Figure A2. (a) The heatmap illustrates the knowledge acquired by the RFLM model for images in the recurrent and non-recurrent classes. Red regions indicate a higher level of attention, while blue regions indicate a lower level of attention. (b) The Euclidean distance plots depict the distance between the common and adaptive features of the four central data. The left side represents common features, while the right side represents adaptive features.

Figure A3. Correlation heatmap of recurrence common features and adaptive features.

Question: Small sample size from four centers limits robustness assessment. The authors should provide more discussions about this issue to clarify the potential concerns. The class imbalance between recurrent and non-recurrent classes may affect federated learning convergence. This issue is very important for clinical application, but this work did not provide comprehensive evaluations or discussions.

Response:

Thank you for the valuable comments from the reviewer. We agreed that during the model construction, small sample size and imbalance ratio of positive and negative samples may bring about negative influence on the model performance. Imbalanced sample ratio can significantly impact the model's classification decision curve, the extraction of task-relevant features, ultimately resulting in bias in diagnostic performance. To address the aforementioned issues, this study has made three improvements in the proposed method:

(1) Federated Learning Framework: In federated learning, the privacy of all local data is effectively preserved, as only the parameters of models trained locally on each data center are shared. There is no direct interaction or sharing of local data between different data centers. During the training process, federated learning aggregates the parameters of local models trained on each data center through a central server to create a global model that synthesizes knowledge from all participating data centers. Subsequently, we utilize the global model to guide and optimize the local models, introducing enhancements that allow the global model to better adapt to the specifics of each local data set. This approach significantly improves the diagnostic performance of the local models on their respective local data. As a result, in this study, federated learning effectively addresses issues related to sample distribution (e.g., small sample size and imbalance sample ratio), while simultaneously extracting common features relevant to the diagnostic task and adaptive features tailored to the characteristics of local data.

(2) Adaptive Feature Extraction: During the process of adaptive feature extraction, the study used a WGAN to generate representative data and built a correlation matrix to reflect domain-specific information among different centers. Subsequently, the use of GCN allows us to learn the correlations between different local models from various centers and integrate this information into inter-domain knowledge. This approach enhances the robustness of the global model, facilitating the construction of a robust model. (*Manuscript: RFLM model exhibits strong robustness*)

(3) Sample imbalance can lead to bias during model optimization. To further mitigate this issue, we also incorporated Focal Loss [4] as the loss function. Focal Loss introduces a weight matrix during model optimization, which assigns a larger weight to the class with fewer data samples when it is misclassified, and a smaller weight to the class with more data samples, thereby balancing the loss between the two classes during optimization. By doing so, the model places greater emphasis on the minority class during training, thereby addressing the sample imbalance and potentially improving the

overall performance of the model. Our algorithm achieved AUC results of 0.710, 0.798, 0.750, and 0.869 on the test sets in four different data centers. (*Manuscript: Construction of RFLM, last Paragraph; Discussion, Paragraph 6*)

Question: *If possible, the authors can provide more ablation studies to analyze the contribution of each sub-component in the proposed method.*

Response:

Thank you for the valuable comments from the reviewer. In our follow-up experiments, we conducted ablation studies on the components of RFLM except for the GAN component, as GAN relies on the GCN component to personalize the algorithm.

Tabel A2 AUC result table of RFLM ablation experiment

CBAM	FED	GCN	GAN	Cohort	Center A	Center B	Center C	Center D
	√	√		Train	0.7445	0.8469	0.8034	0.8529
				Test	0.6503	0.7866	0.7080	0.7818
	√	√	√	Train	0.7190	0.7829	0.8620	0.9412
				Test	0.6542	0.7090	0.7236	0.8073
√	√	√		Train	0.7514	0.7783	0.8047	0.8015
				Test	0.6875	0.6852	0.7194	0.7618
√	√	√	√	Train	0.7502	0.8135	0.8112	0.8750
				Test	0.7101	0.7981	0.8091	0.8691

In the ablation experiments, Group 2 and Group 4 showed that the spatial attention mechanism introduced by CBAM in the Resnet18 network can better extract spatial information of lesions, thereby improving the diagnostic performance of the model. Group 3 and Group 4 experiments proved that using GCN networks to learn domain-specific information from different datasets and incorporating this domain-specific information into personalized strategies can effectively improve the diagnostic performance across all four centers. In the ablation experiments, when GAN components were not used, the GCN network took Euclidean distances between the parameters of each center's model as the personalized input matrix. (*Manuscript: RFLM ablation experiment*)

Question: *How many radiologists defined the region of interest (ROI)? For gastric cancer, it is not easy to accurately identify the tumor area for some cases, how did they deal with that?*

Response:

Thanks for your thoughtful suggestion.

In this study, the ROI (Region of Interest) used was delineated with precision by a radiologist with over 10 years of experience in abdominal imaging diagnosis. This process involved accurately outlining the contours of the lesion. Subsequently, a rectangular frame was constructed based on the exact contour of the lesion, encompassing the entire lesion boundary. As a result, the selection of the ROI is less susceptible to the subjective experiences of clinicians. Additionally, like [5,6], deep learning possesses the capability to automatically extract regions of interest (ROI), eliminating the need for precise manual delineation of ROI areas.

(Manuscript: ROI acquisition)

Question: *4,449 federated radiomics features are extracted in the paper, what are these features?*

Response:

Thank you for the valuable comments from the reviewer.

For a deep learning model, it is an end-to-end system and is typically perceived as a black box, within which features are learned automatically by the model. These features represent abstract representations of the data. In this context, the 4449 features we mentioned are derived from an analysis of each convolutional kernel. Our focus is on extracting specific information from these kernels in the images, enabling model visualization and interpretation. Thus, during this process, we carefully select the most task-relevant features from these 4449 options for further assessment and analysis.

Additionally, these 4449 features can be primarily divided into two categories: shallow features and deep features. Shallow features encompass elements such as the image's contour information and texture details, typically providing insights into the fundamental characteristics of the image. While deep features represent high-dimensional abstract attributes, often directly associated with task discrimination. It's important to clarify that the concepts of shallow and deep features originate from the research conducted by Krizhevsky A and other researchers [7]. In our research, we also conducted visualization of these features, further confirming that the extracted features are indeed task-relevant. *(Manuscript: Figure 8(a))*. Figure S3 in the attachment visualizes the feature extraction process for a patient. *(Manuscript: Construction of RFLM, last Paragraph)*.

References

1. de Liaño AD, Yarnoz C, Aguilar R, Artieda C, Ortiz H. Surgical treatment of recurrent gastric cancer. *Gastric Cancer*. 2008;11(1):10-4. doi: 10.1007/s10120-007-0444-5.

2. Spolverato G, Ejaz A, Kim Y, et al. Rates and patterns of recurrence after curative intent resection for gastric cancer: a United States multi-institutional analysis. *J Am Coll Surg*. 2014 Oct;219(4):664-75. doi: 10.1016/j.jamcollsurg.2014.03.062.
3. Danielsson P E. Euclidean distance map[J]. *Computer Graphics and image processing*, 1980, 14(3): 227-248.
4. Lin TY, Goyal P, Girshick R, He K, Dollar P. Focal Loss for Dense Object Detection. *IEEE Trans Pattern Anal Mach Intell*. 2020 Feb;42(2):318-327. doi: 10.1109/TPAMI.2018.2858826. Epub 2018 Jul 23. PMID: 30040631.
5. Prakash N B, Murugappan M, Hemalakshmi G R, et al. Deep transfer learning for COVID-19 detection and infection localization with superpixel based segmentation[J]. *Sustainable cities and society*, 2021, 75: 103252.
6. Dong S, Yang Q, Fu Y, et al. RCoNet: Deformable mutual information maximization and high-order uncertainty-aware learning for robust COVID-19 detection[J]. *IEEE Transactions on Neural Networks and Learning Systems*, 2021, 32(8): 3401-3411.
7. Krizhevsky A, Sutskever I, Hinton G E. Imagenet classification with deep convolutional neural networks[J]. *Advances in neural information processing systems*, 2012, 25.

Reviewer #2: *This manuscript shows impressive results on a cross-silo federated learning problem of predicting gastric cancer recurrence. The results are promising, but I believe the manuscript needs to be improved and clarified in several respects.*

Response:

Thank you very much for your attention and constructive feedback on our manuscript.

Question: *The organization of the paper makes it somewhat difficult to read. For example, there is much discussion of "features" before the details of the model have been introduced to actually define the features. Too many different adjectives are used to describe features (e.g., "common features", "distinct features", "individual features", "common knowledge/information", "personality information", "federated group learning features", "robust features"). These need to be consolidated and precisely defined before being discussed. Similarly, the discussion in the "Result" section is difficult to read before the details of the machine learning model and setting are provided.*

Response:

We thank you for pointing out this important issue. In order to enhance the readability, we have made two improvements to the manuscript.

First, we have restructured the article in the following order: Introduction, Methodology, Results, and Discussion.

Second, we have made consistent revisions to some key terms throughout the manuscript. For the description of features, common features and common knowledge/information are collectively referred to as common features. It is defined in this manuscript as a feature that is highly correlated between different data centers and has a significant role in the task. The distinct features, individual features, personality information and robust features are collectively referred to as adaptive features. It is defined in this manuscript as a feature within the data center that is highly correlated and has a significant role in the task. The correlation used in this study is Pearson correlation coefficient. The federated group learning features is changed to federated learning radiomic features. It expresses 4449 features extracted from a robust model of each center. Additionally, the federated learning clients and federated learning data centers are collectively referred to as federated learning data centers.

Question: *More details are also needed in order to be able to evaluate the privacy properties of the proposed approach. Following [1, Sec 1.2], it is recommended to include pseudo-code that makes it clear exactly what information is exchanged between the centers and the central server. In particular, it has been well-established that deep networks can memorize their training data (See for example [2],*

as well as many other works on private GANS), and so the use of the WGAN without additional techniques like differential privacy may not be sufficient to preserve privacy.

Response:

We agree with you that the deep networks such as GANS can memorize their training data. We have also taken this into account and taken some steps in building the federal framework.

The parameters of WGAN are not uploaded to the central server. The representative dataset in the study is constructed by training two WGAN networks on two categories (Recurrent group and non-recurrent group) of local data at each data center. Subsequently, we have also taken into consideration that uploading WGAN-generated data could potentially lead to the leakage of local raw data. To address this concern, we superimpose a random Gaussian perturbation matrix on the WGAN-generated data, thereby ensuring the protection of the privacy of the local data. Finally, the generated image with the addition of the disturbance that are called representative dataset is uploaded to a central server for further analysis, while the parameters of the WGAN networks remain stored locally at each data center. We have moved the content of this section from the supplement to such and such position in the revise manuscript (*Construction of RFLM, Paragraph 2*).

Additionally, to further clarify the process of information exchange between different datasets, we have included the pseudocode as follows: Here, n and j represent the optimization constraints, C is the number of GCN iterations, T is the number of local model training iterations, A is the adjacency matrix, and M is the cross-correlation matrix learned by the local model from the representative dataset, n is the number of centers, Y is the model structure, and x represents the x_j^{fake} image in the data.

(*Manuscript: Construction of RFLM, Paragraph 4*).

Algorithm 1 Robust federated based on GCN

- 1: Initialize $\lambda, \gamma, \eta, S, T, C, A, v_i$
 - 2: for each communication round $s=0, 1, \dots, S$ do
 - 3: **Local Update**
 - 4: for local epoch $t=0, 1, \dots, T$ do
 - 5:
$$v_i^{t+1} \leftarrow v_i^t - \eta \nabla (F_i(v_i^t) + \lambda [R(v_i^t, \omega^t) + R(v_i^t, u_i^t)])$$
 - 6: end for
 - 7: **Global Update**
-

-
- 8: $\omega^t \leftarrow \frac{\sum_{i=1}^n v_i}{n}$
- 9: $P_{i,j}^s \leftarrow \text{Splice}(v_1, \dots, v_n)$
- 10: **Update of adjacency matrix A**
- 11: $A_{i,j} = \sum_{i=1}^n \sum_{j=1}^h Y(v_i, x_j^{\text{fake}})$
- 12: **GCN Update**
- 13: for GCN epoch $c=0, 1, \dots, C$ do
- 14: $A^{c+1} \leftarrow A^c - \frac{1}{n} \sum_{i=1}^n (A^c - M^c)$
- 15: end for
- 16: **Robust of model**
- 17: $P_{i,j}^{s+1} \leftarrow \sum_{i=1}^n \sum_{j=1}^k A_{i,j}^s P_{i,j}^s$
- 18: $u^{s+1} \leftarrow P_i^s$
- 19: end for
-

Question: *The relationship between RFLM and standard federated learning algorithms like FedAvg is not clear. FedAvg is not a specific ML model, but rather a model parameter optimization algorithm that can be applied to an arbitrary model architecture in the federated learning setting. However, RFLM appears to imply a particular set of model architectures (including feature extractors and a final classifier), as well as a particular recipe for training those models. If the RFLM algorithm's improvements over other FL methods are a major contribution of the paper that can be applied to other FL tasks, then detailed evaluation on public benchmark datasets should be included to support this point.*

Response:

In this study, RFLM is an improvement upon standard federated learning algorithms. Like FedAvg, RFLM trains local models during the training phase and then aggregates local model parameters at the central server to create a global model. However, what sets RFLM apart is its post-global model refinement process. After obtaining the global model, the local model of each center predicts the

representative data sets of the four centers respectively to obtain the soft label. These soft labels form a $4 \times n$ local model relationship matrix, which represents the inter-domain information between the centers, where 4 is the number of centers and n is the number of all representative data for the 4 centers. Subsequently, based on the acquired local model relationship matrix, GCN (Graph Convolutional Networks) is employed to generate a robust weight parameter. This robust weight parameter captures the topological relationships among the data from each center. The robust weight parameter is combined with each local model to obtain robust local models. The robust local model of each center is the model used to extract features from each center. These improvements allow the model to learn from the unique characteristics and patterns present in each client's data. This helps reduce the non-IID (non-Independently and Identically Distributed) issues caused by different data centers, ensuring that robust local models better match their respective local datasets. Meanwhile, within the RFLM model, to gain a deeper understanding of the robust model and reduce redundancy in the features of local robust models, the study incorporates steps for feature extraction and classification of the local models. The algorithm workflow of RFLM is shown in Figure 2 of the revise manuscript.

Additionally, to demonstrate the applicability of the proposed RFLM algorithm to other tasks, we identified a dataset collected by the Lung Image Database Consortium (LIDC-IDRI) for lung cancer diagnosis, which can be accessed at: <https://wiki.cancerimagingarchive.net/display/Public/LIDC-IDRI>.

LIDC data came from 7 research institutions and 8 medical imaging companies, with a total of 1018 cases. Among them, the malignancy of lung nodules $\geq 3\text{mm}$ is classified into 1-5 grades, with grade 3 being indeterminate malignancy [1]. In this study, lesions with a malignancy of 1-2 grades were classified as benign, while those with a malignancy of 4-5 grades were classified as malignant. Finally, 1746 lesions were included in the dataset. Due to the removal of medical information from each center in the LIDC dataset, we have divided the entire LIDC dataset into four groups, labeled A to D, using a random grouping approach to evaluate the performance of the RFLM algorithm.

We then apply RFLM to the LIDC dataset. The LIDC data distribution is shown in Table A3. The diagnostic performance is shown in Table A4. The AUC for the four central test sets is between 0.811 and 0.852. The effectiveness of the RFLM algorithm proposed in this study was further validated through the results obtained on the publicly available LIDC dataset. (*Manuscript: Research on other tasks*).

Figure 2. RFLM algorithm diagram. (a) The construction process of robust local model in RFLM. (b) Details of generating robust weight parameters in robust local model. (c) Feature extraction and feature classification in RFLM.

Table A3. LIDC multi-center data distribution

Set	Label (n)	Center A	Center B	Center C	Center D
Train	Benign lesion	153	145	182	152

Test	Malignant lesion	101	118	123	95
	Benign lesion	97	124	75	127
	Malignant lesion	72	66	49	67

Table A4. LIDC Multi-center diagnostic performance table

Method	Evaluation	Center A	Center B	Center C	Center D
RFLM (Train)	AUC	0.865	0.842	0.868	0.831
	Sensitive	0.732	0.761	0.781	0.776
		(52/71)	(67/88)	(64/82)	(52/67)
	Specificity	0.907	0.833	0.806	0.778
		(166/183)	(145/174)	(179/222)	(140/180)
	Accuracy	0.858	0.809	0.799	0.777
		(218/254)	(212/262)	(243/304)	(192/247)
RFLM (Test)	PPV	0.754	0.698	0.598	0.565
	NPV	(52/69)	(67/96)	(64/107)	(52/92)
		0.897	0.874	0.909	0.903
RFLM (Test)	AUC	(166/185)	(145/166)	(179/197)	(140/155)
		0.816	0.811	0.852	0.824
	Sensitive	0.681	0.650	0.807	0.778
		(32/47)	(26/40)	(25/31)	(28/36)
	Specificity	0.803	0.812	0.763	0.684
		(98/122)	(121/149)	(71/93)	(108/158)
	Accuracy	0.796	0.778	0.774	0.701
(130/169)		(147/189)	(96/124)	(136/194)	
RFLM (Test)	PPV	0.571	0.482	0.532	0.359
	NPV	(32/56)	(26/54)	(25/47)	(28/78)
		0.867	0.896	0.922	0.931
		(98/113)	(121/135)	(71/77)	(108/116)

Question: Tables 1-2 and Figures 1-2 seem to use a single fixed train/test split as given in Table 3. However, given the relatively small number of examples, using 4 significant digits seems unjustified for these metrics. Further, rather than reporting the metrics for a single train/test split, it would be preferable to use cross-validation, and train multiple models on different train/test splits so the mean metrics could be reported together with appropriate confidence intervals. Fig 3 (b) appears to partially address this. However, the exact experiment is unclear; "five random permutations" in particular doesn't make sense to me, as a permutation refers to a re-ordering of the data, but Fig 3(a) clearly shows re-assignment of the examples to different train/test splits. However, in the 5 distributions, the ratio of train/test seems to vary significantly --- for cross validation, I would have expected a fixed train/test ratio, and just different partitions. I would suggest removing Fig 3 and simply using cross-validation to provide confidence intervals in Tables 1-2 and Figures 1-2, which would free up space to

improve clarity on the other points I have raised.

Response:

We thank you for pointing out this important issue. To present the results more effectively, we have standardized all results in the paper to three significant figures.

In the experimental section, we assessed the model's resistance to interference by employing a method that involved five random shufflings of data distributions. This approach allows us to evaluate how the overall results change when the model encounters different data distributions, thereby assessing the model's robustness. At your suggestion, in our subsequent experiments, we introduced a cross-validation component. Given the limited amount of data in individual data centers, we utilized a three-fold cross-validation approach. The specific results are depicted in Figure A4. The average AUC results for cross-validation across the four data centers test sets were 0.722, 0.774, 0.755, 0.813, respectively. The experimental results demonstrated that the permutation of datasets had minimal impact on the performance of the multi-center model. (*Manuscript: RFLM model exhibits strong robustness, Paragraph 2*).

Figure A4. Four center three-fold cross-validation ROC curves. The blue curve is the AUC average of the three-fold curve ROC. The gray area is the upper and lower limits of the ROC curve.

Question: *It is unclear to me how the subjective CT findings (line 294) are used. Are these five CT signs (line 303) then used as features in the RFLM model, or does the model only use the ROI from the CT images as the raw input features for prediction? Or were these signals used in the clinical model?*

Response:

In this study, the clinical model relies entirely on subjective CT features and pathological information. The RFLM model utilizes image data only from the regions of interest (ROIs) delineated and selected by radiologists.

When building the clinical model, given that clinical models typically rely on statistically based modeling, different data centers may incorporate different clinical indicators when constructing clinical models, which could hinder result validation. Therefore, for the clinical model, we identified clinically significant features by referencing relevant literature [2,3] and selected three independent risk factors (CA199, N, M stage) from the five CT features and pathological information. Subsequently, we employed a random forest strategy to build the clinical model using multicenter data.

In order to create unnecessary misunderstanding, we included these subjective signs and pathological information in the attached introduction to building clinical models. (***Supplementary: S5, Subjective CT findings and pathological evaluations were conducted.***)

Question: *Finally, the future of this work is unclear. While the results are promising, I would like to see a more robust discussion of what would be necessary to see these models actually deployed in a clinical setting. For example, do the models do well when applied to a center where they were not trained? This could be testing by a 4-way cross validating, training on 3 of the centers and evaluating on the final held-out center. Does the model still work if a different radiologist selects the ROIs for the training data vs the test data?*

Thank you for the valuable comments from the reviewer. To assess the generalization capabilities of RFLM, we conducted two sets of experiments. (1) we trained the model on three different data centers and evaluated its performance on a separate, independent data center. We repeated this process four times in a cross-validation framework to examine how well RFLM performs on data centers that were not part of the training phase. (2) we trained a traditional deep learning model on a single dataset and tested it on the other three data centers. This was done to demonstrate the superiority of RFLM compared to traditional deep learning algorithms. It's important to note that RFLM generates distinct robust models for different datasets. Therefore, when validating on a separate data center, we needed to select the most suitable robust model from those used during training for feature extraction. Our chosen strategy involved calculating the Euclidean distance between the feature representations of

each data center used in training and the individual validation data center. We then selected the robust model with the closest distance to serve as the feature extraction model for the separate validation data center.

The results of experiment (1) and (2) are shown in **Table A5 and A6**, respectively. The results of two sets of experiments indicate that the RFLM algorithm continues to perform well on data centers that have not been subjected to its training. This may be attributed to the algorithm's ability to extract shared features, which play a significant role in diagnosing R-AGC and NR-AGC. However, data centers that have not undergone RFLM training lack these adaptive features, leading to relatively inferior results compared to those that participated in RFLM training.

Additionally, in this study, the ROI delineation strategy employed involved the precise outlining of lesion contours by radiologists, followed by enclosing the entire lesion within a rectangular bounding box. As a result, the selection of ROIs is less susceptible to subjective differences among radiologists.

Table A5 Rotate and cross-validate experimental results

Center A	Center B	Center C	Center D	AUC of the validation data center test set	Robust model used
√	√	√	★	0.7727	Center A
√	√	★	√	0.7251	Center B
√	★	√	√	0.7504	Center C
★	√	√	√	0.7073	Center B

Where √ is the data center involved in training, ★ is the data center used for verification, and Robust model used is a robust model used for feature extraction of data centers used for verification.

Table A6 Multicenter independent validation of AUC result tables

Center A training, Center B, C, D validation				
Center A (Train)	Center A (Test)	Center B	Center C	Center D
0.7412	0.6740	0.5053	0.5342	0.7382
Center B training, Center A, C, D validation				
Center B (Train)	Center B (Test)	Center A	Center C	Center D
0.7690	0.7178	0.5128	0.5057	0.5218
Center C training, Center A, B, D validation				
Center C (Train)	Center C (Test)	Center A	Center B	Center D
0.8073	0.6923	0.5681	0.6455	0.5109
Center D training, Center A, B, C validation				
Center D (Train)	Center D (Test)	Center A	Center B	Center C
0.8088	0.7055	0.5215	0.5423	0.5755

Question: *"without exposing data privacy" should perhaps be "exposing private data". But more importantly, the authors should clarify more explicitly exactly what the privacy benefit is. For example, can they make statements like "raw CT images are never shared between centers; instead model parameters (including a generative WGAN trained on the CT images) are exchanged."*

Response:

Thank you for the valuable comments from the reviewer. We have revised the introduction as follows:

In recent years, Artificial intelligence (AI) technology has received widespread attention in the medical field and has shown exciting results [11,12,13]. However, a stable and effective AI-assisted diagnostic model not only on appropriate algorithms, but also on large training dataset [14,15]. This large training dataset requires patient data to be shared across medical centers. At this time, medical organizations to relinquish control of their own data. The security and privacy of patient data will be difficult to protect, especially between countries, and even create a data monopoly situation. Therefore, multi-center data sharing is difficult to achieve in reality.

Some researchers have proposed federated learning (FL). FL is to train a single shared model on a center by aggregating local models that is trained using only its own data from each medical center. FL ensures data security while encouraging multi-center collaboration and could lead to the development of more accurate and general AI-assisted diagnostic system [16]. However, due to the differences of medical image collection equipment and regions in various medical centers, multi-center data may have problems of non-independent identically distributed (such as different ratio of positive and negative data samples and different distribution of image data) [17,18,19]. This will make the shared model in federated learning unable to meet the needs of all centers, such as the performance of the shared model in the A-center is better, and the performance of the shared model in the B-center is worse.

Driven by these real-world issues, we have developed a personalized Federated Learning Model (PFLM) in this study to accurately predict the risk of postoperative recurrence in patients with AGC. By effectively combining raw CT image data from multiple centers while ensuring the privacy of individual patients, our model significantly enhances predictive performance and generality without relying on centralized control over the final model. Moreover, in order to validate the effectiveness of the RFLM model, this paper examines the characteristics of RFLM. It utilizes Pearson correlation analysis to identify shared features among different data centers and the robust characteristics of each individual data center. (*Manuscript: Introduction*).

At the same time, we have also supplemented the discussion section with further advantages of privacy protection. RFLM algorithm ensures data privacy by not sharing the original CT images

between data centers, relying solely on model parameters, including those trained on CT images (generated through WGAN). This approach minimizes the disparity between the global model and the robust model to the fullest extent. (*Manuscript: Discussion, Paragraph 6*).

Question: line 143: How is "correlation" between features computed? It would help to know the mathematical type of the features, and give a precise formula if possible.

Response:

Thank you for the valuable comments from the reviewer. The correlation used in this study is Pearson correlation coefficient [20]. (*Manuscript: Assessment of common and individual features of RFLM*). The specific formula is as follows:

$$R = \frac{\sum((X_i - \tilde{y}) \times (Y_i - \tilde{y}))}{\sqrt{\sum(X_i - \tilde{y})^2 \times \sum(Y_i - \tilde{y})^2}} \quad (1)$$

Where: R represents the Pearson correlation coefficient. X_i represents the i -th observation of variable X . Y_i represents the i -th observation of variable Y . \tilde{y} represents the mean (average) of variable X . \tilde{y} represents the mean (average) of variable Y . When $R = 1$, it indicates a perfect positive correlation. When $R = -1$, it indicates a perfect negative correlation. When $R = 0$, it indicates no linear relationship. The closer the value of the Pearson correlation coefficient is to 1 or -1, the stronger the linear relationship between the two variables, while a value close to 0 indicates little to no linear relationship.

Question: line 168: Unclear what "attention" refers to

Response:

Thank you for the valuable comments from the reviewer. In this paper, the term "attention" refers to the model's focus or level of interest in different regions of an image. Red regions indicate a high level of attention, while blue regions signify a lower level of attention. (*Manuscript: Figure 8, notes*).

Question: "inter-group / intra-group" -> it should be clarified exactly what "group" refers to. I believe this is either R or NR (the label / class in typical machine learning terminology), but that isn't completely clear, as it could be confused with site.

Response:

Thank you for the valuable comments from the reviewer. To avoid noun confusion, we have replaced the terms "inter-group" and "intra-group" in the text with "between each data center" and

"within each data center," respectively.

Question: line 242: "two tasks" -> what are the tasks? does this mean the groups R and NR? Again, use consistent terminology and define it clearly early on, for example "we consider two groups (classes, labels) throughout, NR (no-recurrent) and R (recurrent)"

Response:

Thank you for the valuable comments from the reviewer. "two tasks" refer to the two categories of tasks, namely NR-AGC and R-AGC. In order to avoid ambiguity, the two tasks were revised to identify the recurrence and non-recurrence of advanced gastric cancer in the revised manuscript. (*Manuscript: Discussion, paragraph 6*).

Question: line 323: "client models" The paper has used "center" consistently to refer to the four medical centers. I believe "client" here is a synonym for "center". While "client" is often used in the FL literature, it would be more clear to pick one term and stick to it throughout the paper, remarking on the connection to other terminology when first introduced.

Response:

Thank you for the valuable comments from the reviewer. All terms representing federated learning centers in the text have been uniformly changed to "center" to avoid any noun confusion.

Question: line 352: What is the type of these features? For example, is each feature value a single number? a vector or matrix of a specific size? To make this concrete, please clarify.

Response:

Thank you for the valuable comments from the reviewer. In this study, the convolution kernel of each central robust model was used as a feature extractor to extract multiple feature maps from the local CT image data of each patient, and then the mean value of each feature map was calculated as a joint radiomics feature. Since the central robust model has a total of 4449 convolution cores, so a total of 4,449 federated radiomic features were extracted for one patient. The features extracted in this paper form a matrix of dimensions $n \times 4449$, where each row represents an individual patient, and each column represents a specific feature. Each individual feature within the matrix is represented as a numerical value. (*Manuscript: Construction of RFLM, paragraph 5*).

Question: line 373: It's unclear how these 4 groups of 200 features relate to previous discussion for example in Fig. 4 and Fig. 5. What are Fig. 4's A_1 , ..., A_4 ?

Response:

Thank you for the valuable comments from the reviewer.

To explore the common and individual features within the RFLM model, this paper employed U-tests and mRMR (Minimum Redundancy Maximum Relevance) to select the top 200 most valuable features in each data center for analysis. For the sake of clarity in presenting the results, Pearson correlation coefficients were used to compute the similarity between different center features and features within the local data center. This approach allowed us to identify the top 5 features with the highest within each data center correlations as adaptive features and the features with the highest between each data center correlations as common features within each data center. A_1 ,..., A_5 in Figure 4 (a) represents the 5 common features selected from center A. A_1 ,..., A_5 in Figure 4 (b) represents the five adaptive features selected from center A.

Figure 5b shows the Euclidean distance between the common feature and the adaptive feature between the two centers. This visualization indicates that the similarity among common features is greater than that among individual features.

References

1. MCNITT-GRAY M F, MEYER C R, et al. The Lung Image Database Consortium (LIDC) data collection process for nodule detection and annotation[J]. *Academic Radiology*, 2007, 14 (12) :1464-1474.
2. Chongqing T, Liubao P, Xiaohui Z, et al. Cost–Utility Analysis of the Newly Recommended Adjuvant Chemotherapy for Resectable Gastric Cancer Patients in the 2011 Chinese National Comprehensive Cancer Network (NCCN) Clinical Practice Guidelines in Oncology: Gastric Cancer. *PharmacoEconomics*. 2014;32(3):235-243. doi:10.1007/s40273-013-0065-2.
3. Kodera Y, Yamamura Y, Shimizu Y, et al. The Number of Metastatic Lymph Nodes: A Promising Prognostic Determinant for Gastric Carcinoma in the Latest Edition of the TNM Classification. *J Am Coll Surg*. 1998;187(6):597-603. doi:10.1016/S1072- 7515(98)00229-4.
4. Sung Hyuna, Ferlay Jacques, Siegel Rebecca L, et al. Global Cancer Statistics 2020: GLOBOCAN Estimates of Incidence and Mortality Worldwide for 36 Cancers in 185 Countries. *CA-A CANCER JOURNAL FOR CLINICIANS*. 2021;71(3):209-249. doi:10.3322/caac.21660.
5. Smyth Elizabeth C, Nilsson Magnus, Grabsch Heike I, et al. Gastric cancer. *LANCET*. 2020;396 (10251):635-648. doi:10.1016/S0140-6736(20)31288-5.
6. Jiang Yuming, Li Tuanjie, Liang Xiaoling, et al. Association of Adjuvant Chemotherapy with

- Survival in Patients with Stage II or III Gastric Cancer. *JAMA Surgery*. 2017;152(7): e171087. doi:10.1001/jamasurg.2017.1087.
7. Noh Sung Hoon, Park Sook Ryun, Yang Han-Kwang, et al. Adjuvant capecitabine plus oxaliplatin for gastric cancer after D2 gastrectomy (CLASSIC): 5-year follow-up of an open-label, randomised phase 3 trial. *LANCET ONCOLOGY*. 2014;15(12):1389-96. doi:10.1016/S1470-2045(14)70473-5.
 8. Japanese gastric cancer treatment guidelines 2018 (5th edition). *Gastric Cancer*. 2021;24 (1):1-21. doi:10.1007/s10120-020-01042-y.
 9. Gambardella Valentina, Cervantes Andrés. Precision medicine in the adjuvant treatment of gastric cancer. *LANCET ONCOLOGY*. 2018;19(5):583-584. doi:10.1016/S1470-2045(18)30131-1.
 10. Chen Dexin, Fu Meiting, Chi Liangjie, et al. Prognostic and predictive value of a pathomics signature in gastric cancer. *Nature communications*. 2022;13(1):6903. doi:10.1038/s41467-022-34703-w.
 11. Fu Yu, Jung Alexander W, Torne Ramon Viñas, et al. Pan-cancer computational histopathology reveals mutations, tumor composition and prognosis. *Nature cancer*. 2020;1(8):800-810. doi:10.1038/s43018-020-0085-8.
 12. Sirinukunwattana Korsuk, Domingo Enric, Richman Susan D, et al. Image-based consensus molecular subtype (imCMS) classification of colorectal cancer using deep learning. *GUT*. 2021;70 (3):544-554. doi:10.1136/gutjnl-2019-319866.
 13. Schmauch Benoît, Romagnoni Alberto, Pronier Elodie, et al. A deep learning model to predict RNA-Seq expression of tumours from whole slide images. *Nature communications*. 2020;11 (1):3877. doi:10.1038/s41467-020-17678-4.
 14. Echle Amelie, Grabsch Heike Irmgard, Quirke Philip, et al. Clinical-Grade Detection of Microsatellite Instability in Colorectal Tumors by Deep Learning. *Gastroenterology*. 2020;159 (4):1406-1416.e11. doi:10.1053/j.gastro.2020.06.021.
 15. Campanella Gabriele, Hanna Matthew G, Geneslaw Luke, et al. Clinical-grade computational pathology using weakly supervised deep learning on whole slide images. *Nature medicine* .2019;25 (8):1301-1309. doi:10.1038/s41591-019-0508-1
 16. Alireza Fallah, Aryan Mokhtari, and Asuman Ozdaglar. Personalized federated learning: A meta-learning approach. In *NeurIPS*, 2020. arXiv preprint arXiv:2002.07948 (2020).

17. T Dinh, Canh, Nguyen Tran, and Josh Nguyen. " Personalized federated learning with moreau envelopes." *Advances in Neural Information Processing Systems* 33 (2020): 21394-21405.
18. McMahan, Brendan, et al. "Communication-efficient learning of deep networks from decentralized data." *Artificial intelligence and statistics*. PMLR, 2017.
19. Li, Tian, Anit Kumar Sahu, Manzil Zaheer, et al. "Federated optimization in heterogeneous networks." *Proceedings of Machine learning and systems* 2 (2020): 429-450.
20. Cohen I, Huang Y, Chen J, et al. Pearson correlation coefficient[J]. *Noise reduction in speech processing*, 2009: 1-4.

REVIEWERS' COMMENTS

Reviewer #1 (Remarks to the Author):

The authors have answered the questions I was concerned about.

Reviewer #3 (Remarks to the Author):

This article introduces a robust federated learning method for identifying high-risk patients. The proposed approach is reasonable and easy to understand, and the results show notable improvements over many existing methods.

Since the authors have already made revisions, some concerns have been raised by peer reviewers and the authors have addressed them. Nonetheless, I hope the authors can further clarify the following points:

- 1) The authors add discussions on the privacy protection of generated data. However, this part is relatively weak because no theoretical guarantee is provided. The authors can follow the differential privacy framework and calculate the privacy budget for reference if possible.
- 2) The medical insights in this work are relatively weak. The authors explore interpreting the model features, while how these results can guide clinical diagnosis and operations is unclear. Furthermore, since the predictions in many centers are still not accurate enough, the authors need to discuss the potential ethical risks of model usage.
- 3) It seems that model generality is not well verified. It is not guaranteed that the proposed method consistently works well on different model backbones and the interpretation results are consistent.

Responses to Reviewer Comments

Reviewer #3: *This article introduces a robust federated learning method for identifying high-risk patients. The proposed approach is reasonable and easy to understand, and the results show notable improvements over many existing methods. Since the authors have already made revisions, some concerns have been raised by peer reviewers and the authors have addressed them.*

Response:

First of all, we thank you for your constructive comments. Your helpful suggestions have greatly improved our current work.

Question: *The authors add discussions on the privacy protection of generated data. However, this part is relatively weak because no theoretical guarantee is provided. The authors can follow the differential privacy framework and calculate the privacy budget for reference if possible.*

Response:

Thank you for providing valuable feedback! Data privacy protection is indeed a crucial concern. Differential privacy techniques achieve data protection by introducing random noise into the local original data, perturbing the raw data locally to ensure privacy. The corresponding privacy budget is obtained by calculating the ratio of data sensitivity to noise scale [1]. However, in our study, there was no interaction with original data; instead, we utilized representative data generated by a WGAN network for interactions.

The representative dataset in this study was constructed by training two WGAN networks on two types of data (recurrent and non-recurrent groups) in each data center. Subsequently, we also considered the possibility of local raw data leakage due to the upload of WGAN-generated data. To address this concern, we applied a random Gaussian perturbation matrix to the WGAN-generated data, ensuring privacy protection for local data. Finally, the noisy representative dataset was uploaded to the central server for further analysis.

To validate the privacy protection capability of our algorithm on the original data, we employed deep learning networks to classify both the original and the noisy representative datasets. If the algorithm can distinguish between these two classes of images, we infer significant dissimilarity between them. The classification experiment results are presented in Table A1. The results indicate that both the VGG16 and ResNet18 networks can effectively differentiate between these two datasets. Therefore, we believe there is a significant difference between the two datasets, and uploading the noisy representative data to the central server for analysis will not result in the leakage of original data.

Table A1 Classification accuracy of raw data and representative data sets with added noise

Model	Set	Center A	Center B	Center C	Center D
Resnet18	Train	0.999	0.993	0.998	0.988
	Test	0.993	0.997	0.999	0.987
VGG16	Train	0.997	0.996	0.997	0.987
	Test	1.000	1.000	1.000	1.000

Question: *The medical insights in this work are relatively weak. The authors explore interpreting the model features, while how these results can guide clinical diagnosis and operations is unclear. Furthermore, since the predictions in many centers are still not accurate enough, the authors need to discuss the potential ethical risks of model usage.*

Response:

In clinical practice, surgical resection remains the primary treatment method for advanced gastric cancer patients. However, the relatively high postoperative recurrence rate leads to suboptimal survival outcomes. Therefore, we have developed an auxiliary diagnostic model, RFLM, to quantitatively assess the high-risk patients prone to recurrence after curative gastric resection. This model assists clinicians in tailoring personalized treatments to improve patient prognosis.

Due to variations in data collection equipment and data quality across different medical centers, the performance of artificial intelligence-assisted diagnostic models is often suboptimal when deployed in various different data center [2]. Therefore, to address the heterogeneity issue in deep learning diagnostic models for multicenter data, we developed an RFLM algorithm. The algorithm was tested on multicenter data related to postoperative recurrence in advanced gastric cancer, demonstrating promising performance. Additionally, to validate the effectiveness of the algorithm, we conducted a parallel study on pulmonary nodule data from the publicly available LIDC dataset [3]. As shown in Table A2, our algorithm demonstrated similarly excellent performance. This indicates that the results of our algorithm present a promising biomarker in assisting clinical doctors with diagnosis.

Given that the deep learning techniques employed in our study represent a black box, meaning the decision-making process of the model is not transparent, we aimed to enhance the interpretability of the algorithm. Following methodologies outlined in relevant studies [4,5], we conducted class activation map analysis, as illustrated in Figure A1a, and corresponding feature analysis (Figure A1b). These analyses provide visualizations of the model's decision-making process, contributing to a better understanding of its inner workings.

Figure A1. (a) The heatmap illustrates the knowledge acquired by the RFLM model for images in the recurrent and non-recurrent classes. Red regions indicate a higher level of attention, while blue regions indicate a lower level of attention. (b) The Euclidean distance plots depict the distance between the common and adaptive features of the four central data. The left side represents common features, while the right side represents adaptive features.

Table A2. LIDC Multi-center diagnostic performance table

Method	Evaluation	Center A	Center B	Center C	Center D
RFLM (Train)	AUC	0.865	0.842	0.868	0.831
	Sensitive	0.732	0.761	0.781	0.776
		(52/71)	(67/88)	(64/82)	(52/67)
	Specificity	0.907	0.833	0.806	0.778
		(166/183)	(145/174)	(179/222)	(140/180)
	Accuracy	0.858	0.809	0.799	0.777
		(218/254)	(212/262)	(243/304)	(192/247)
PPV	0.754	0.698	0.598	0.565	
(52/69)	(67/96)	(64/107)	(52/92)		
NPV	0.897	0.874	0.909	0.903	
	(166/185)	(145/166)	(179/197)	(140/155)	
RFLM (Test)	AUC	0.816	0.811	0.852	0.824
	Sensitive	0.681	0.650	0.807	0.778
		(32/47)	(26/40)	(25/31)	(28/36)
	Specificity	0.803	0.812	0.763	0.684
		(98/122)	(121/149)	(71/93)	(108/158)
Accuracy	0.796	0.778	0.774	0.701	
(130/169)	(147/189)	(96/124)	(136/194)		

PPV	0.571 (32/56)	0.482 (26/54)	0.532 (25/47)	0.359 (28/78)
NPV	0.867 (98/113)	0.896 (121/135)	0.922 (71/77)	0.931 (108/116)

Question: *It seems that model generality is not well verified. It is not guaranteed that the proposed method consistently works well on different model backbones and the interpretation results are consistent.*

Response:

To validate the generalizability of our research model, we replaced the original backbone network in the model framework with VGG16 [6] and conducted experiments. The results, as shown in Table A3, indicate that the model maintains comparable performance after replacing the backbone network. This suggests that the proposed model in our study exhibits relatively consistent effectiveness across different backbone networks.

Table A3 The RFLM algorithm diagnoses performance in four data centers on the VGG16 network

Method	Evaluation	Center A	Center B	Center C	Center D
RFLM (VGG16) (Train)	AUC	0.743	0.802	0.820	0.890
	Sensitive	0.588 (30/51)	0.773 (17/22)	0.875 (21/24)	1.000 (4/4)
	Specificity	0.815 (106/130)	0.735 (36/49)	0.688 (22/32)	0.677 (23/34)
	Accuracy	0.751 (136/181)	0.747 (53/71)	0.768 (43/56)	0.711 (27/38)
	PPV	0.556 (30/54)	0.557 (17/30)	0.677 (21/31)	0.267 (4/15)
	NPV	0.835 (106/127)	0.878 (36/41)	0.880 (22/25)	1.000 (23/23)
	RFLM (VGG16) (Test)	AUC	0.713	0.766	0.775
Sensitive		0.500 (20/40)	0.593 (16/27)	0.769 (20/26)	0.909 (10/11)
Specificity		0.778 (56/72)	0.762 (32/42)	0.704 (19/27)	0.700 (35/50)
Accuracy		0.679 (76/112)	0.696 (48/69)	0.736 (39/53)	0.738 (45/61)
PPV		0.556 (20/36)	0.615 (16/26)	0.741 (20/28)	0.400 (10/25)
NPV		0.737 (56/76)	0.744 (32/43)	0.760 (19/25)	0.972 (35/36)
RFLM		AUC	0.750	0.814	0.811

(Resnet18 CBAM) (Train)	Sensitive	0.824 (42/51)	0.636 (14/22)	0.667 (16/24)	1.000 (4/4)
	Specificity	0.562 (73/130)	0.918 (45/49)	0.875 (28/32)	0.647 (22/34)
	Accuracy	0.635 (115/181)	0.831 (59/71)	0.786 (44/56)	0.684 (26/38)
	PPV	0.424 (42/99)	0.778 (14/18)	0.800 (16/20)	0.250 (4/16)
	NPV	0.890 (73/82)	0.849 (45/53)	0.778 (28/36)	1.000 (22/22)
	AUC	0.710	0.798	0.809	0.869
RFLM (Resnet18 CBAM) (Test)	Sensitive	0.700 (28/40)	0.482 (13/27)	0.731 (19/26)	0.909 (10/11)
	Specificity	0.542 (39/72)	0.857 (36/42)	0.778 (21/27)	0.640 (32/50)
	Accuracy	0.598 (67/112)	0.710 (49/69)	0.755 (40/53)	0.689 (42/61)
	PPV	0.459 (28/61)	0.684 (13/19)	0.760 (19/25)	0.357 (10/28)
	NPV	0.765 (39/51)	0.720 (36/50)	0.750 (21/28)	0.970 (32/33)

References

1. Bebensee B. Local differential privacy: a tutorial[J]. arXiv preprint arXiv:1907.11908, 2019.
2. Linardos A, Kushibar K, Walsh S, et al. Federated learning for multi-center imaging diagnostics: a simulation study in cardiovascular disease[J]. Scientific Reports, 2022, 12(1): 3551.
3. MCNITT-GRAY M F, MEYER C R, et al. The Lung Image Database Consortium (LIDC) data collection process for nodule detection and annotation[J]. Academic Radiology, 2007, 14 (12) :1464-1474.
4. Xu HL, Gong TT, Liu FH, Chen HY, Xiao Q, Hou Y, Huang Y, Sun HZ, Shi Y, Gao S, Lou Y, Chang Q, Zhao YH, Gao QL, Wu QJ. Artificial intelligence performance in image-based ovarian cancer identification: A systematic review and meta-analysis. EClinicalMedicine. 2022 Sep 17;53:101662. doi: 10.1016/j.eclinm.2022.101662. PMID: 36147628; PMCID: PMC9486055.
5. Tang Z, Chuang K V, DeCarli C, et al. Interpretable classification of Alzheimer's disease pathologies with a convolutional neural network pipeline[J]. Nature communications, 2019, 10(1): 2173. <https://www.nature.com/articles/s41467-019-10212-1>.
6. Simonyan K, Zisserman A. Very deep convolutional networks for large-scale image recognition[J].

arXiv-preprint arXiv:1409.1556, 2014.